# Towards Machine Theory of Mind with Large Language Model-Augmented Inverse Planning

## Abstract

We propose a hybrid approach to machine Theory of Mind (ToM) that uses large language models (LLMs) as a mechanism for generating hypotheses and likelihood functions with a Bayesian inverse planning model that computes posterior probabilities for an agent's likely mental states given its actions. Bayesian inverse planning models can accurately predict human reasoning on a variety of ToM tasks, but these models are constrained in their ability to scale these predictions to scenarios with a large number of possible hypotheses and actions. Conversely, LLM-based approaches have recently demonstrated promise in solving ToM benchmarks, but can exhibit brittleness and failures on reasoning tasks even when they pass otherwise structurally identical versions. By combining these two methods, this approach leverages the strengths of each component, closely matching optimal results on a task inspired by prior inverse planning models and improving performance relative to models that utilize LLMs alone or with chain-of-thought prompting, even with smaller LLMs that typically perform poorly on ToM tasks. We also exhibit the model's potential to predict mental states on open-ended tasks, offering a promising direction for future development of ToM models and the creation of socially intelligent generative agents.

## 1 Introduction

The capacity for Theory of Mind (ToM)—the ability to infer the beliefs, desires, intentions, and goals, of others—is a hallmark of human social cognition, underpinning humans' ability for social interaction, communication, and collaboration.

Interdisciplinary work at the intersection of cognitive science and computer science (e.g., Jara-Ettinger, 2019; Baker et al., 2017; Langley et al., 2022; Rabinowitz et al., 2018) has aimed to both characterize the mechanisms that make our ability to understand other minds so powerful, and leverage our understanding of humans' ToM to design machines with a comparable capacity for social inference. These insights are critical for the design of trustworthy social agents that can be relied upon to align their understanding of situations with those of humans (Street, 2024).

Nevertheless, efforts to develop robust machine ToM have faced substantial challenges. On one hand, Bayesian models of cognition inspired by inverse reinforcement learning have offered a promising computational framework for human reasoning on a variety of ToM tasks (e.g., Baker et al., 2017; Ullman et al., 2009; Jara-Ettinger et al., 2016; Jara-Ettinger, 2019), but like many Bayesian models, face challenges with implementation outside of environments with heavily restricted hypothesis and action spaces. On the other hand, recent work with large language models (LLMs) has argued that their success on a variety of benchmarks represents a significant advancement in the development of machine ToM (e.g., Kosinski, 2024; Gandhi et al., 2024); however, the extent to which these successes represent robust and general social reasoning abilities is unclear, as several analyses have revealed that the performance of LLMs on tasks outside of existing benchmarks is often brittle (Trott et al., 2023; Shapira et al., 2023; Ullman, 2023), and alignment of LLMs with human reasoning in open-ended domains remains elusive (Amirizaniani et al., 2024).

In this paper, we present a hybrid approach, LLM-Augmented Inverse Planning (LAIP), that exploits the potential complementary strengths of Bayesian inverse planning models and LLMs.

By integrating the generative capabilities of LLMs, inverse planning models can be theoretically unbounded in the quantity of hypotheses about an agent's beliefs and desires, or actions given the agent's state, that they can entertain in any given situation. On the other hand, by explicitly formalizing the process of inverse planning, we show this hybrid model is less susceptible to zero-shot reasoning errors than LLMs without specific prompting or with generic chain-of-thought (CoT) prompting.

## 2 RELATED WORK

**Theory of Mind in Humans**   ToM is a foundational component of human social cognition. It emerges early in childhood (Wimmer & Perner, 1983; Gopnik & Astington, 1988; Wellman et al., 2001), possibly even in infancy (Butterfill & Apperly, 2013), allowing human beings to make sophisticated inferences about how others' beliefs, desires, and knowledge may differ from one's own. By allowing people to infer when others do not know what we do—and when others know what we do not—theory of mind has been proposed as a necessary component to the extent and breadth of human systems of cooperation and trust, and thus indirectly the success of human culture (Frith & Frith, 2010; Gopnik & Meltzoff, 1993; Tomasello et al., 1993).

**Bayesian Inverse Planning as Theory of Mind**   Inspired by probabilistic models of human cognition (e.g., Chater et al., 2006; Tenenbaum & Griffiths, 2001, work by Verma & Rao (2005), Baker et al. (2009; 2011), and Rafferty et al. (2015) formalized the understanding of others' beliefs, desires, and intentions as an instance of Bayesian reasoning within a partially observable Markov decision process (POMDP). Within this framework, an observer engages in *inverse planning*—inverting the observer's own process of generating an action policy based on its beliefs and desires—in order to reason about the unobserved internal states that give rise to an agent's behaviours. These models have been extended to account for both children's and adults' commonsense reasoning that others will act according to a naive form of expected utility, maximizing expected rewards and minimizing costs (Jara-Ettinger et al., 2016; 2020; Lucas et al., 2014). Within this broader framework, ToM can be thought of as equivalent to inverse reinforcement learning (IRL; Jara-Ettinger, 2019; Ruiz-Serra & Harré, 2023), recovering an agent's reward structure from actions that are assumed to be generated by an optimal policy given the agent's beliefs. A family of models have extended this framework to various MDP and POMDP settings (Lim et al., 2020; Wei et al., 2023; Wu et al., 2023).

**Deep Learning models of Theory of Mind**   Deep learning methods have proven tremendously successful at learning complex strategies that reach or surpass human ability across a variety of complex games (Mnih et al., 2015; Silver et al., 2018). Rabinowitz et al. (2018) developed an early deep learning model for ToM based on meta-learning: by learning to predict several disparate classes of agents that have different preferences and action policies, the model can extrapolate an agent's likely policy after observing only a few actions. Other models have explored other aspects of ToM reasoning: for example, including explicit belief models improves the performance of agents on cooperative and adversarial games in multiagent settings (Fuchs et al., 2021; Moreno et al., 2021; Oguntola et al., 2023; Wen et al., 2019). However, these methods have encountered challenges in their ability to successfully capture ToM (Aru et al., 2023)—in particular, the fact that deep learning algorithms may implement "shortcuts" to solve theory of mind tasks.

**Theory of Mind in LLMs**   Most recently, the emergence of powerful, generally capable LLMs has led to investigation of their capacity for ToM. Earlier investigations found that LLMs scored substantially below human level on ToM tasks (Sap et al., 2022), although performance has rapidly increased with the release of newer models (Bubeck et al., 2023; Gandhi et al., 2024; Kosinski, 2024). However, a recurring concern with these results is the robustness and generalizability of these successes. For example, Ullman (2023); Shapira et al. (2023) showed that small alterations to ToM tasks can drastically decrease the rate of correct responses, cautioning that LLMs' success on ToM benchmarks may reflect a successful deployment of heuristics and shortcuts, much like deep learning models, and that this success may not generalize to broader task settings.

To this end, several works have investigated the degree to which additional prompts or model components could successfully guide LLMs to more robust performance on ToM tasks. In the same vein as chain-of-thought (CoT) prompting (Wei et al., 2022) improved LLMs' performance on various questions, such as mathematics and symbolic reasoning problems, Zhou et al. (2023) showed that LLMs struggled on ToM scenarios that focused on "thinking for doing" (i.e., making choices for its

own interventions on the world based on reasoning about others' knowledge states), and suggested that prompts focusing on imagining future states and reflecting on the model's ability to intervene in these scenarios can improve ToM-consistent choices in these scenarios. A similar concept used by Wilf et al. (2023) that prompts LLMs to take the perspective of the target agent also exhibits higher accuracy on false-belief questions. Li et al. (2023) prompts LLM-based generative agents to maintain and update an explicit belief state based on the environment, finding this improves the performance of the agents on a collaborative task. Sclar et al. (2023) decomposes ToM questions into a symbolic graphical representation, simplifying the task complexity for the LLM. Some models (Cross et al., 2024; Shi et al., 2024; Zhang et al., 2024; Zhi-Xuan et al., 2024) have also extended investigations of using LLMs as components of ToM evaluation in multi-agent settings, allowing for the comparison of one agent's evaluations of another agent's beliefs and goals to be compared against a ground truth.

## 3 LLM-AUGMENTED INVERSE PLANNING

A promising avenue for Theory of Mind comes from augmenting inverse planning with LLMs (Zhi-Xuan et al., 2024; Jin et al., 2024). When provided with hypotheses, LLM-augmented agents can reason across these hypotheses, using multimodal information and generate appropriate mental inferences about a social partner or an observed target. We extend this line of work by designing an explicit inverse planning model that uses an LLM to generate hypotheses and consider the likelihoods of possible actions across a potentially open-ended hypothesis and action space. Thus, we aim to design a model for machine ToM that is more robust to the identified shortcomings of both traditional Bayesian models as well as LLMs. An important advantage of utilizing LLMs in this hybrid approach is their ability to sidestep the frame problem in Theory of Mind reasoning (Shanahan, 1997). In traditional Bayesian models, researchers are often required to manually define the hypothesis space in advance, constraining the system's understanding of possible mental states and the ways these states might be updated by candidate actions (Dennett, 1987). However, extensively pre-trained LLMs can sample hypotheses implicitly from its representations of language and world knowledge, generating plausible candidate hypotheses for a given scenario that can accommodate more open-ended environments.

An overview of the architecture of the LAIP model is presented in Figure 1 (see also Algorithm 1 in Appendix A.1). Broadly, the model conducts Bayesian inverse planning to reason about a target agent's preferences given its action. After first generating a prior belief over possible hypotheses regarding the agent's preferences, the LLM observes the agent's situation and its observation of the environment at each timestep of a task. Then, the LLM simulates the agent's perspective on the task, generating reasoning about the agent's likely choices given the state. From this reasoning, it generates the likelihood of different possible actions under each of these hypotheses. After the agent acts, the LLM updates the posterior distribution over hypotheses given the action chosen by the agent.

## 4 EXPERIMENTS

To evaluate the LAIP model, we first adapt a series of experiments inspired by the scenario described in Baker et al. (2011), in which a Bayesian ToM model and humans observed differing agent trajectories in an environment with partial visibility, in which food trucks were sometimes present and absent, and had to reason about the agent's underlying food preferences and beliefs about whether an option was available.

Within our task, LLMs similarly observe an agent moving between different options of restaurants, and must infer the agents' beliefs about whether a restaurant that is not visible to them is open or closed, as well as their preferences for different foods based on their actions in the environment. This environment provides an initial test of the capacity of the LAIP model that can be explicitly compared against optimal models. In these studies, we restrict the action space (Studies 1 and 2) and the hypothesis space (Study 2) in order to make it possible to compare these models to the predictions of a Bayes-optimal model. In Study 3, we explore the model's hypothesis and action generation capabilities explicitly.

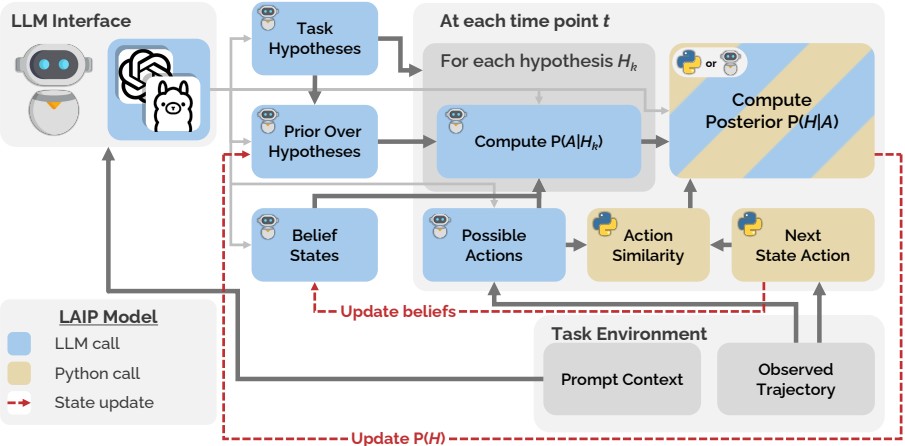

Figure 1: Schematic of LAIP model. The LLM generates candidate hypotheses and prior probability of each hypothesis, as well as beliefs over task-relevant states for the actor. Then, the LLM generates actions according to the current state in the trajectory, which are used to compute the likelihood of actions given each hypothesis. Finally, the generated action is compared to the next state's action, and a posterior is computed either mathematically or within the LLM.

## 4.1 RESTAURANTS TASK

In Studies 1 and 2, we present LLMs with an environment in which they must move between a series of rooms to visit one of three restaurants (Figure 2). Restaurants may be open or closed, but the agent does not know whether a restaurant is open unless the room containing the restaurant is visible. From any given room, only some of the other rooms in the environment are visible, so if an agent has not yet visited a room from which a given restaurant is visible, the agent does not know whether a restaurant is open or closed. Agents begin in a room where only the first restaurant (Chinese) is visible, but after moving into other rooms, are able to find out whether the Mexican restaurant or Japanese restaurant is open.

## 4.2 STUDY 1

In Study 1, we focus on an agent trajectory in one of two possible world states. In both world states, both the Chinese and Mexican restaurant are open; however, the Japanese restaurant is open in one world and closed in the other world. The agent moves from Room 1, to Room 2, to Room 3, and then back to Room 2, and finally to the Chinese restaurant.

When the Japanese restaurant is closed, the agent's actions are consistent with a strong preference for the Japanese restaurant, followed by the Chinese restaurant, followed by the Mexican restaurant. Since the agent is not able to observe whether the Japanese restaurant is open until reaching Room 3, the agent moving away from the Japanese restaurant after observing that it is closed does not have a bearing on the agent's perceived preferences, while the fact that it moves towards the Chinese restaurant afterward indicates a preference for the Chinese restaurant over the Mexican restaurant. However, when the Japanese restaurant is open, the agent's actions are not consistent with any strong preference hierarchy, and may reflect weak or inconsistent preferences. Thus, a model reasoning about the agent's preferences based on its actions and a representation of the agent's belief states should infer a strong preference for the Japanese restaurant based on the agent's policy.

### 4.2.1 EXPERIMENTAL DESIGN AND PROCEDURE

In Study 1, we compare the performance of the LAIP model to a model with a generic prompt to directly infer the posterior distribution given the situation and the agent's actions (zero-shot baseline). Both models used GPT-4o (OpenAI, 2023) to generate their responses, and received a common list of 20 candidate hypotheses about the agent's preferences for the different restaurants, also generated by GPT-4o. In the main text, we present the results with a uniform prior over hypotheses, but we additionally present model results using LLM-generated prior beliefs in Appendix **??**. We completed 10 runs per model per trajectory.

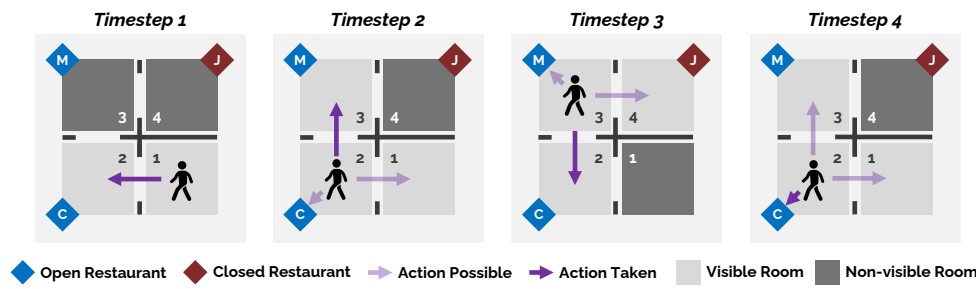

Figure 2: Schematic of task design and observed trajectory for Study 1. The observed actor moves between rooms. At each timestep, the actor chooses whether to move to a new room or eat at a restaurant in the same room.

At each timestep, the models received a system prompt containing information about the environment, including all rooms, all restaurants, all legal movement paths between rooms, and rooms from which each restaurant is visible. Additionally, the prompt stated that restaurants were almost always open, but were sometimes closed, and that agents could not eat a closed restaurant. In addition to the system prompt, each LLM call contained information about the agent's current room, the visibility from the current room, including any visible restaurants and whether they are open or closed, and the rooms connected to the current room that an agent can move to.

### 4.2.2 RESULTS

Our main analysis of interest in Study 1 was whether the models would 1) successfully infer the target agent's preference for Japanese food when the Japanese restaurant was closed, and 2) infer a weaker or inconsistent preference when the Japanese restaurant was open. Based on the hypotheses generated by the LLM, we identified one hypothesis ($H_2$: The agent prefers Japanese food the most, then Chinese, then Mexican) that would be most consistent with the agent's preferences when the Japanese restaurant was closed. Although the agent's choices are less strongly diagnostic when the Japanese restaurant is open, we also identified three hypotheses that would be more strongly compatible with the agent's actions ($H_9$: The agent will choose at random; $H_{18}$: The agent would choose between Chinese and Japanese depending on its plans after lunch; $H_{20}$: The agent is not particularly picky, but will likely choose the most convenient option, Chinese).

We operationalized model performance in two ways. First, we compared the proportion of the posterior distribution placed on these hypotheses in each condition across different model settings,

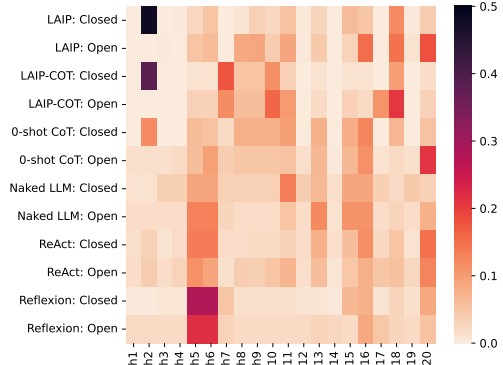

Figure 3: Posterior probabilities for hypotheses after the final timestep when the Japanese restaurant is open or closed. Darker colours indicate higher posterior probability of hypotheses (columns). When the Japanese restaurant is closed (odd rows), only the LAIP model infers that the agent's actions are most consistent with a preference for the Japanese restaurant, followed by the Chinese restaurant, followed by the Mexican restaurant ($H_2$).

including the full LAIP model, the LAIP model using a single prompt as a chain-of-thought, a zero-shot baseline, and two baselines: ReAct (Dagan et al., 2023) and Reflexion (Shinn et al., 2024), iterative reasoning and reflection-based m odels of decision-making. Further, we predict that the model prediction should sharply diverge between timesteps 2 and 3, since this action is most strongly diagnostic of the agent's preferences, reflecting the point where a rational observer would observe the strongest change in beliefs based on the agent's actions, once the agent has observed whether each restaurant is open or closed.

Overall, our findings suggest that the LAIP model is able to effectively combine LLM inputs and inverse planning to infer the agent's likely preferences given its actions. When the Japanese restaurant was closed, the LAIP model gave a posterior probability of 48.4% to $H_2$, compared to 11.9% for the zero-shot CoT prompt, 3.7% for ReAct, 0.3% for Reflexion, and 1.2% for the zero-shot baseline (Figure 3). Conversely, when the Japanese truck was open, the LAIP model assigned just 0.3% probability to this hypothesis, compared to 1.9% for the zero-shot baseline.

The posterior distribution is more diffuse for the LAIP model when the Japanese restaurant is open, but assigns 41.8% of its posterior probability to one of $H_9$, $H_{18}$, or $H_{20}$. The zero-shot baseline, by contrast, assigns 12.6% probability to the combination of these three choices, while ReAct assigns 21.7%, Reflexion assigns 10.4%, and the zero-shot CoT assigns 28.6% probability, respectively. Across both conditions, ReAct and Reflexion instead became more confident that the target agent preferred Chinese food (its ultimate choice, but inconsistent with its initial move towards the Japanese restaurant).

Moreover, we observed the divergences in the LAIP model's posterior probability between timesteps by computing the Hellinger distance and the Jensen-Shannon divergence (JSD; a symmetrized form of the KL divergence), three measures of similarity between probability distributions, to assess the distance between the prior and posterior distributions at each timestep. As we predicted, the strongest divergence occured between timesteps 2 and 3 (Open: $H(P,Q) = 0.445$, $\text{JSD}(P||Q) = 0.169$, Closed: $H(P,Q) = 0.47$, $\text{JSD}(P||Q) = 0.191$), moreso than for the next closest difference between timesteps, 1 and 2 (Open: $H(P,Q) = 0.297$, $\text{JSD}(P||Q) = 0.075$, Closed: $H(P,Q) = 0.303$, $\text{JSD}(P||Q) = 0.076$). This suggests that the model's endorsement of hypotheses changed most when the agent's actions most strongly indicated a preference.

### 4.3 STUDY 2

In Study 1, we showed that the LAIP model is capable of generating hypotheses that capture different possible agent preferences, reasoning about the likeliest actions taken by agents given those possibilities, and utilizing inverse planning to draw appropriate ToM-consistent conclusions from the agent's actions. To show our model's robustness across tasks as well as across LLMs of differing sizes, and enable comparison with an optimal model inspired by BToM, we employ a common set of hypotheses across all models and task setups.

#### 4.3.1 EXPERIMENTAL DESIGN AND PROCEDURE

**Model Configurations** We assessed six distinct model configurations. Three configurations include varying amounts of the LAIP model algorithm, while two represent a basic CoT baseline and zero-shot LLM baseline. For comparison, we also include an optimal model that uses Bayesian inference to infer the agent's preferences.

*LAIP (Full model):* This model follows each step as laid out in Algorithm 1, except that we constrain the action state. At each step, the model is presented with the the agent's state, and then simulates the likelihood of the agent taking an action given each hypothesis being true with a separate LLM call. After generating the likelihoods, we normalize action probabilities to sum to 1 and compute the posterior probability mathematically using Bayes' rule.

*LAIP (LLM computes posterior):* This model is identical to the Full Model, except the computation of the posterior is done through an LLM call instead of mathematically. The prompt provides the LLM with the prior probability of all hypotheses as well as a matrix representing the probability of all actions given each candidate hypothesis. Then, it stated the action chosen, and asked the LLM to compute the posterior probabilities of all hypotheses.

*LAIP (Single CoT):* This model is instructed to perform all of the actions of the LAIP model in order to compute the posterior distribution; however, this is done using a single LLM call, rather than a separate call for each potential hypothesis.

*Generic CoT:* The model is presented with the agent's situation, the hypotheses, and the prior probability of each hypothesis being true. After being presented with the agent's action, the LLM is then asked to compute the posterior probability for each hypothesis, and finally instructed to think step by step (e.g., Wei et al., 2022).

***Zero-Shot Baseline:*** This model omits the instruction to think step by step, but is otherwise identical to the Generic CoT.

***Optimal Model:*** This model does not use an LLM, and uses Bayesian inference to analytically compute the agent's likely preferences given its actions. This model assumes an agent starts with a baseline belief of $P(\text{open}) = 0.95$ for all restaurants, and it will move towards its most preferred restaurant using the most efficient path with a probability of $P(\text{open})(1 - \varepsilon)$, with a probability of $\varepsilon = 0.01$ that the agent will move to a random room. When a restaurant becomes visible to an agent, the agent will update its beliefs of $P(\text{open})$ to 1 or 0, depending on whether it is open or closed.

**LLMs used**   We used GPT-3.5, GPT-4o, GPT-4o-mini (OpenAI, 2023), Mixtral (Jiang et al., 2024), LLaMA 3-70B, LLaMA 3-8B (Dubey et al., 2024), and Gemma 2 (Gemma Team, 2024) as the LLMs for generating likelihoods. All LLMs were used for all LLM model configurations with the exception of Gemma 2, which did not provide posterior probabilities for the LAIP (Single CoT), Generic CoT, and zero-shot baseline conditions. We completed 5 runs per trajectory per model configuration-LLM pair.

**Experimental Design**   Study 2 employed the same environment as Study 1 (Figure 2). However, we tested ten different agent trajectories, each of which is compatible with a different set of preference hierarchies on the part of the agent. Trajectories 1 and 9 correspond to the Japanese Closed and Japanese Open trajectories used in Study 1, respectively. Each trajectory varied which restaurants were open or which restaurant an agent moved towards, which should lead to different inferences about the agent's preferences. A full list of trajectory details is found in Appendix A.1.

### 4.3.2   RESULTS

Overall, we find that across different agent trajectories, different LLMs, and different measures, the LAIP models consistently outperform the Generic CoT and the zero-shot baseline models. The LAIP models exhibit higher accuracy, higher correlation to the predictions of the optimal model (Table 2), lower distance metrics (Table 4), and assign more probability to hypotheses supported by the agent's actions (Figure 4).

**Alignment with Correct Forward Models**   For Trajectories 1–8, the agents' trajectories are compatible with between one and three forward models. Thus, a model's inferences about an agent's preferences are correct to the extent that they align with these plausible hypotheses. In Figure 4, we show the performance across the average of these probabilities per trajectory. Overall, we find that only the full LAIP model results in inferences that are consistently above the predictions of a uniform distribution, while the LAIP model with the LLM-computed posterior also performs well with larger models (GPT-4o, GPT-4o mini, and LLaMA 3-70B), consistent with an advantage in mathematical reasoning for larger LLMs (e.g., Yuan et al., 2023).

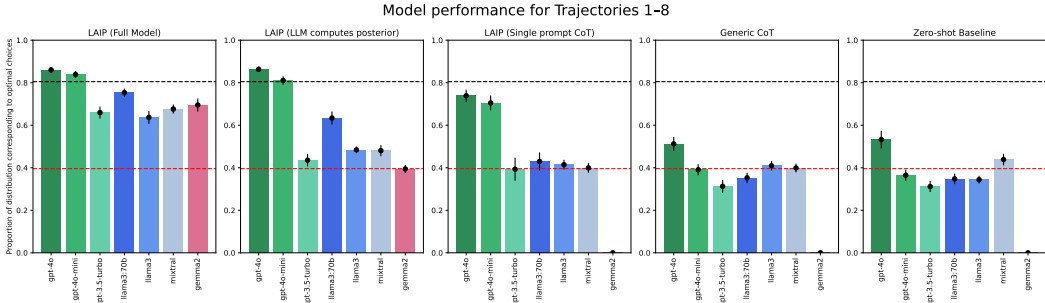

Figure 4: Empirical results for LLMs using each model configuration, averaged across Trajectories 1–8. Bars indicate the proportion of posterior distribution assigned to the options that correspond to the options considered most probable by the optimal model. Red dashed line indicates the expected outcome for a uniformly random distribution across all hypotheses. Black dashed line indicates optimal model. Gemma 2 was not run for LAIP (Single CoT), Generic CoT, and Zero-shot Baseline.

**Correlations**     The LAIP models also exhibited consistently high correlations with the optimal model. The full model was the best performing for 6 out of 7 LLMs, and was significantly correlated with the optimal model in all 7 (all $r \geq .546$, $p < .001$; all $\rho \geq .519$, $p < .001$; see Appendix A.4 for full results). Only GPT-4o, GPT-4o mini, and LLaMA 3-70B were significantly correlated with the optimal model when the LLM computed the posterior distribution, and only GPT-4o and GPT-4o mini were correlated with the Single CoT version of the LAIP model. Notably, no baseline models produced results significantly correlated with the optimal model. These results suggest that even models that otherwise demonstrated lower performance on the task, such as LLaMA 3-8B, were still performing substantially more accurately using the LAIP model with the mathematically computed posterior than they were any other conditions.

While we do not have direct human performance data for this specific tasks, we note the very strong correlation between the LAIP model and the optimal model ($r = .94$ for GPT-4o). Given that similar tasks by Baker et al. (2011) showed similarly strong correlations between their Bayesian ToM model and human responses, this suggests that the LLMs' choices on our tasks closely match human intuitions.

**Distance Metrics**

Across 6 of 7 LLMs, LAIP with the mathematically computed posterior distribution has the lowest or tied for the lowest Jensen-Shannon divergence (see Appendix A.4 for full results). For the remaining models (GPT-4o), LAIP with the LLM-computed posterior distribution had the lowest distance, followed by LAIP with the mathematically computed posterior distribution; however, GPT-4o's performance was excellent overall, outperforming all other models in each study.

**Effect of Model Size on Efficacy of LAIP**

We observed that the LAIP model, particularly when the posterior was mathematically computed rather than computed using the LLM, exhibited larger improvements for smaller models relative to larger ones. Indeed, we found that the average improvement on the full model relative to the LLM computes posterior model tends to increase with smaller models, and shows no difference for the largest model (GPT-4o: Cohen's $d = -0.03$, $t(14) = -0.05$, $p = .96$; Mixtral: Cohen's $d = 1.10$, $t(14) = 2.19$, $p = .046$; Gemma 2: Cohen's $d = 1.59$, $t(14) = 3.17$, $p = .007$; others non-significant; see Appendix A.4 for full results). This finding suggests that decomposing the tasks involved in ToM into smaller components, and offloading tasks such as mathematical reasoning which smaller, less expressive language models can struggle with, can substantially improve the performance of these models on ToM tasks.

## 4.4 MMToM-QA Performance

To exhibit LAIP's effectiveness at solving tasks with more complex environments, we tested LAIP on the goal inference tasks on the MMToM-QA benchmark (Jin et al., 2024), a series of 300 scenarios involving an individual searching for objects in an apartment. With a larger number of potential goals and broader action space, this serves as a strong test of LAIP's ability to represent an agent's goals given more ambiguous states.

In Table 1, we show the results of our model on the MMToM-QA dataset, comparing its performance both to BIP-ALM (Jin et al., 2024), an inverse-planning model with a fine-tuned LLM and symbolic planner, to Sim-ToM (Wilf et al., 2023), Symbolic-ToM (Sclar et al., 2023), and baseline GPT-4, as well as with human performance on the same dataset. We find that LAIP is very accurate across multiple versions of the goal inference task, and overall displays a higher accuracy than other text models, particularly on the "goal given updated belief" tasks.

## 4.5 Unconstrained Action Spaces

In studies 1 and 2, we demonstrated that LAIP effectively improved ToM reasoning compared to the zero-shot baselines in controlled environments. Yet, the ultimate benefit of LAIP comes from its ability to navigate open-ended environments where inferences need to be made regarding non-obvious mental states. In such situations, multiple hypotheses need to be constructed based on previous experiences as perceives need to determine the relevant cues from the environment. Our extension allows the model to be able to not only generate hypotheses, but also potential actions to

Table 1: Performance on goal inference tasks of MMToM-QA dataset.

| Model | Goal-True | Goal-False | Goal-Updated | Goal-Future | All |
|---|---|---|---|---|---|
| Humans | 85.8 | 76.7 | 65.0 | 68.3 | 74.0 |
| GPT-4 | 48.0 | 42.7 | 2.7 | 42.7 | 34.0 |
| Sim-ToM w/ GPT-4 | 61.3 | 44.0 | 2.7 | 54.7 | 40.7 |
| Symbolic-ToM w/ GPT-4 | 73.3 | 66.7 | 0.0 | 50.7 | 47.7 |
| BIP-ALM w/ GPT-J (text only) | 77.3 | **68.0** | 30.7 | **70.7** | 61.7 |
| LAIP w/ GPT-4 | **78.4** | 46.6 | **80.4** | 64.3 | **67.5** |

evaluate, and dynamically update each of these next step. Here, we extend LAIP to infer mental state attributions in a more realistic and ambiguous scenario.

In this study, we use a scenario involving two coworkers, Carol and Alice. Carol of them is planning a surprise birthday party for Alice, and needs to make reservations a restaurant, but does not know Alice's food preferences (see Appendix A.5. We introduce four individual scenes where Alice (a generative agent using Gemma 2) makes a choice about what to eat. Carol observes the setting for the scene (what kinds of options are available, and common knowledge available to both people) and Alice's concrete actions (what choice to eat), then, using LAIP with GPT-4o, infer the likely preferences given Alice's actions.

The LAIP model generates 20 hypotheses about Alice's preferences based on the initial information about their workplace. Then, at each step, it generates six possible actions to condition on given the state context, and generates the likelihood of the actions conditioned on each hypothesis being true. Then, the model observes Alice's action as a string. Since this action may not line up with the actions generated by LAIP, we compute the cosine similarity of the ground truth observation $O$ to the actions generated by LAIP $A_i \in A$: $S(O, A_i)$, then compute the posterior distribution, using the softmax function to normalize the cosine similarity values:

$$P(H|O) = \text{softmax}(S(O, A_i))P(A|H)P(H)$$

In three of the four timesteps, Alice chooses to eat something that does not match any of her desired foods due to alternative factors (availability, restricted options due to location, illness). Although choosing these foods might otherwise indicate a preference for "plain" or "comfort" foods, because of the context, they are not inconsistent

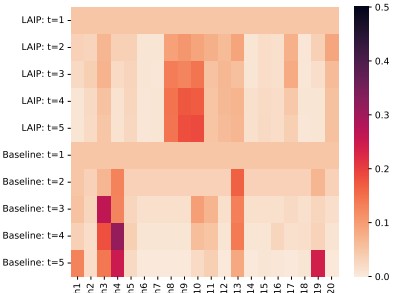

Figure 5: Posterior probabilities for hypotheses for the LAIP (left) and zero-shot baseline (right) models after the final timestep. Darker colours indicate higher posterior probability of hypotheses (columns). LAIP, but not the baseline model, places the highest probability density on Alice's true preferences ($H_9$, $H_{10}$).

with a preference for other foods. As a result, the LAIP model correctly infers that Alice has a preference for Thai and Indian food, inferring two of Alice's true preferences ($P(H \in \{H_9, H_{10}\}) = .371$) much more often than the zero-shot baseline ($P(H \in \{H_9, H_{10}\}) = .047$; see Figure 5). Thus, even though Alice only acts on her preferences once, selecting Thai food, the LAIP model correctly infers that not choosing it in other circumstances does not reflect her preferences for other foods, nor does choosing these other foods reflect strong preferences for these particular foods. The baseline model, on the other hand, infers that Alice prefers to eat "plain" or "comfort" foods more often ($P(H \in \{H_1, H_4, H_{19}\}) = .622$), heavily lowering the probability that Alice would like something else.

By explicitly conditioning the likelihood of actions on possible preferences whose influences on actions may vary according to the situation, LAIP is able to generate not only plausible actions and reason about how observing these actions should change one's beliefs, but also reason about when observing them should *not* change one's beliefs, i.e., when one has little choice but to eat at the only restaurant in a small town, this does not suggest that one has a preference for the food served by that restaurant.

This design could be extended further within interactive environments of generative agents (Park et al., 2023), where hypotheses could be further refined within a social environment. In open-ended environments where agents might communicate their own beliefs, preferences, and goals in idiosyncratic ways, we suggest that the efficacy of generative models at inferring these latent states could be substantially improved through the inclusion of explicit inverse planning algorithms.

## 5  DISCUSSION AND CONCLUSION

Our studies highlight that inverse planning models and large language models can complement each others' strengths. While many existing models of inverse planning are able to capture important elements of human inferences about belief and desire on a variety of ToM-relevant tasks, these models are often constrained by the space of potential hypotheses and actions that are available within any given scenario or task. Conversely, evaluations of LLMs' social reasoning abilities (e.g., Shapira et al., 2023; Ullman, 2023) have emphasized that LLMs can often succeed on these tasks through the use of shallow heuristics that do not generalize to adversarial examples or more complex, ecologically valid situations.

By exploiting the capacity of LLMs to serve as a generative model for hypotheses and actions— in essence, functioning as a theory and action sampler in an unbounded hypothesis space—while using inverse planning to engage in reasoning more similarly to humans in comparable settings, LAIP shows promise as a tool to enable the application of Bayesian models in a broader number of settings. Further, we observed that it was particularly successful in improving the social reasoning abilities of smaller LLMs relative to the baseline, highlighting the efficiency of our architecture and showing the promise of "hybrid" architectures pairing LLMs with other tools such as direct mathematical computation. These findings also

**Limitations and Future Work**

LAIP's potential computational cost mirrors the challenges of human social cognition. Human beings in real-world environments rationally allocate cognitive resources to reasoning according to factors such as motivation and ability, often relying on inexpensive heuristics when the cost of errors is low, and engaging in more effortful processing when necessary (Lin et al., 2010).

In the same vein, people may trade off the benefits of a more accurate epistemic representation against the benefits of a wider hypothesis space (Dasgupta et al., 2017). By setting the number of hypotheses to consider higher or lower, LAIP can similarly represent differing degrees of effort or reflection. While we consider a fixed number of hypotheses, maintaining a lower number of hypotheses and then sequentially revising these hypotheses upon observing evidence in a manner similar to particle filter models (e.g. Sanborn et al., 2010) would enable low-probability hypotheses to be dismissed while maintaining and revising more likely ones. Thus, extensions to this line of work should consider how methods such as sequential Monte Carlo (SMC) can combine importance sampling methods to approximate a posterior distribution and revising hypotheses via proposals drawn from an LLM, which may further optimize the high cost of sampling and evaluating hypotheses, while generating more human-like performance on ToM tasks.

Given the training procedure of LLMs, their use as hypothesis and action samplers has the potential to result in the proposal of biased or stereotypical hypotheses or actions that have the potential to be propagated and entrenched. Differing prior beliefs about what a hypothesis space ought to look like can result in drastically different beliefs—some of which could be negative. Although this can also be true of human reasoning, we urge that care should be taken in interpreting these results as more "rational", particularly in situations where the inferences that may be drawn might be harmful towards marginalized or disadvantaged groups.

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

# A APPENDIX

The following sections contain additional information and measures from Studies 1 and 2.

## A.1 LAIP ALGORITHM

---
**Algorithm 1** The LAIP Model

---
**Ensure:** $T$: total timesteps, $S_1 \dots S_t$: agent state per timestep
1: **while** $t < T$ **do**
2:    **if** $t = 0$ **then**
3:       $P(H) \leftarrow$ generate_llm_prior$(S)$
4:                   $\triangleright$ Generate a prior over possible hypotheses given the world state.
5:    **else**
6:       $P(H) \leftarrow P(H|A_{t-1})$
7:    **end if**
8:    **for** $H_i$ in $H$ **do**
9:       $\{A_1 \dots A_N\} \leftarrow$ generate_llm_actions$(S, H_k)$       $\triangleright$ Reason about the likely actions
10:       **for** $A_j$ in $\{A_1 \dots A_N\}$ **do**
11:          $P(A_j|H_i) \leftarrow$ generate_llm_likelihood$(S)$
12:       **end for**
13:    **end for**
14:    $O \leftarrow$ llm_observe()               $\triangleright$ Observe the agent's ground truth action.
15:    $P(H|O) \propto P(A|O)P(A|H)P(H)$    $\triangleright$ Computed mathematically or via LLM call.
16: **end while**
17: **return** $P(H|O)$

---

## A.2 TRAJECTORY DETAILS FOR STUDIES 1 AND 2

| Trajectory | Actions | | | | Restaurants |
|---|---|---|---|---|---|
| Study 1: Open | Room 2 | Room 3 | Room 2 | Chinese | All open |
| Study 1: Closed | Room 2 | Room 3 | Room 2 | Chinese | Japanese closed |
| 1 | Room 2 | Room 3 | Room 2 | | Japanese closed |
| 2 | Room 2 | Room 3 | Room 4 | | All open |
| 3 | Room 2 | Room 3 | Mexican | | All open |
| 4 | Room 2 | Chinese | | | All open |
| 5 | Room 2 | Room 3 | Room 4 | | Mexican closed |
| 6 | Room 2 | Chinese | | | Mexican closed |
| 7 | Room 2 | Room 3 | Mexican | | Chinese closed |
| 8 | Room 2 | Room 3 | Room 4 | | Chinese closed |
| 9 | Room 2 | Room 3 | Room 2 | | All open |
| 10 | Room 2 | Room 3 | Room 4 | | Chinese/Mex. closed |

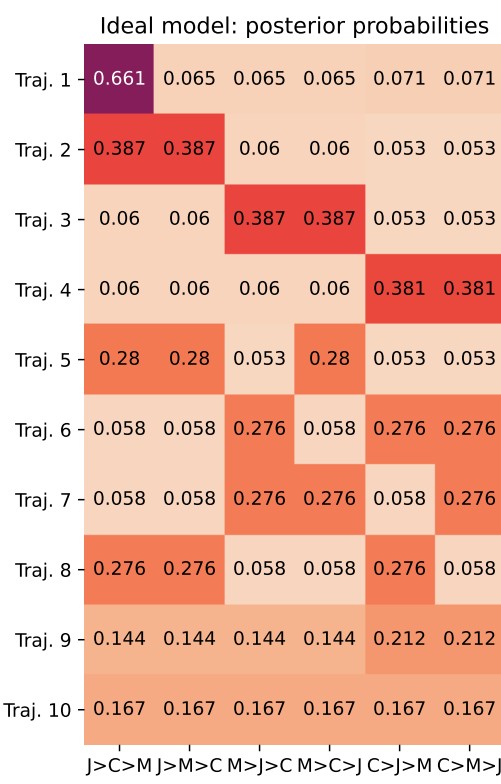

Figure 6: Ideal model results for Trajectories 1 through 10.

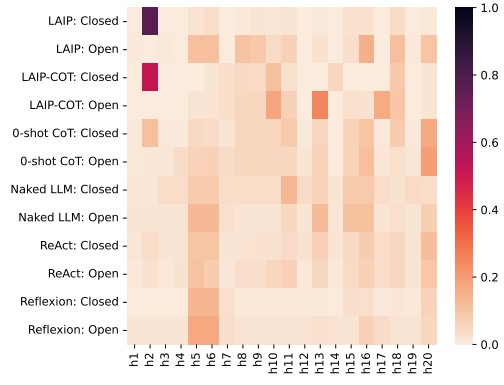

Figure 7: Study 1: Results with generated prior beliefs.

## A.3 STUDY 1 PROMPT

---

**System Prompt:**
You are observing a person's actions and trying to determine how much the person likes three types of food: Japanese, Chinese and Mexican.
This is the set of rules to help you determine the preferences:

There are seven rooms:

- Room 1 connects to Room 2.
- Room 2 connects to Room 1, Room 3, and Room 4.
- Room 3 has a Chinese restaurant in it.
- Room 4 is connected to Room 2, Room 5, and Room 6.
- Room 5 has a Mexican restaurant in it.
- Room 6 connects to Room 4 and Room 7.
- Room 7 has a Japanese restaurant in it.

Restaurants are visible from different rooms:

- The Chinese restaurant is **visible** from Room 1, Room 2, and Room 4.
- The Chinese restaurant is **not visible** from Room 6.
- The Mexican restaurant is **visible** from Room 2, Room 4, and Room 6.
- The Mexican restaurant is **not visible** from Room 1.
- The Japanese restaurant is **visible** from Room 4 and Room 6.
- The Japanese restaurant is **not visible** from Room 1 or Room 2.

The agent knows for sure if a restaurant is open if it gets close enough to it. Each restaurant is almost always open, but sometimes is closed. If the restaurant is closed, it will not open up later. Agents cannot eat at restaurants that are closed, even if they like a food.

**Hypothesis Generation:**
Imagine that Bob is at a food court. There are 3 restaurants, and he thinks that they are likely open, but can't be sure until he gets closer. The options are Japanese food and Mexican food, and Chinese food. Bob will need to walk past the Chinese food to get to the Japanese food and the Mexican food, and will not be able to see if they are actually open until he walks past. Your first task is to consider Bob's food preferences. Think about the options that Bob has and think about which food he likes best, second best, and third best. Write out a series of hypotheses about his preferences. You can add additional hypotheses that you think may influence his opinions. Write out 20 hypotheses, and try to have them as mutually exclusive as possible and things that you think will direct his actions as he walks through the space. Provide a likelihood for the probability of each hypothesis. You will use his actions to determine which hypothesis is most accurate by watching each step, updating your beliefs about how much each is likely true for Bob.

---

Figure 8: Hypothesis generation prompt for Study 1.

A.4 METRICS FOR STUDY 2

Table 2: Correlation coefficients (Pearson $r$) between LLM models and optimal models for probability values for all hypotheses in Trajectories 1–10. **Bold** indicates model(s) with the lowest distance metric for the given LLM.

| Model | LAIP (Full) | LAIP (LCP) | LAIP (CoT) | Generic CoT | Baseline |
| --- | --- | --- | --- | --- | --- |
| GPT-4o | 0.943 | **0.971** | 0.796 | 0.264 | 0.219 |
| GPT-4o mini | **0.960** | 0.923 | 0.611 | -0.028 | -0.099 |
| GPT-3.5 | **0.620** | -0.019 | 0.171 | -0.213 | -0.151 |
| LLaMA 3-70B | **0.742** | 0.521 | 0.127 | -0.087 | -0.099 |
| LLaMA 3-8B | **0.546** | 0.299 | 0.043 | 0.039 | -0.110 |
| Mixtral | **0.639** | 0.242 | 0.038 | 0.017 | 0.056 |
| Gemma 2 | **0.680** | -0.056 | – | – | – |

Table 3: Correlation coefficients (Spearman $\rho$) between LLM models and optimal models for probability values for all hypotheses in Trajectories 1–10. **Bold** indicates model(s) with the lowest distance metric for the given LLM.

| Model | LAIP (Full) | LAIP (LCP) | LAIP (CoT) | Generic CoT | Baseline |
| --- | --- | --- | --- | --- | --- |
| GPT-4o | 0.923 | **0.951** | 0.828 | 0.294 | 0.330 |
| GPT-4o mini | **0.947** | 0.912 | 0.611 | 0.033 | -0.028 |
| GPT-3.5 | **0.544** | 0.054 | -0.097 | -0.065 | -0.144 |
| LLaMA 3-70B | **0.655** | 0.567 | 0.166 | -0.177 | -0.134 |
| LLaMA 3-8B | **0.563** | 0.336 | 0.007 | 0.045 | -0.052 |
| Mixtral | **0.620** | 0.125 | 0.101 | -0.038 | 0.098 |
| Gemma 2 | **0.701** | -0.044 | – | – | – |

Table 4: Jensen-Shannon divergence values between LLM models and optimal models, averaged across Trajectories 1–10, for each model configuration. **Bold** indicates model(s) with the lowest distance metric for the given LLM.

| Model | LAIP (Full) | LAIP (LCP) | LAIP (CoT) | Generic CoT | Baseline |
| --- | --- | --- | --- | --- | --- |
| GPT-4o | 0.015 | **0.011** | 0.042 | 0.109 | 0.112 |
| GPT-4o mini | **0.022** | **0.022** | 0.075 | 0.223 | 0.214 |
| GPT-3.5 | **0.113** | 0.212 | 0.277 | 0.234 | 0.211 |
| LLaMA 3-70B | **0.068** | 0.086 | 0.168 | 0.180 | 0.150 |
| LLaMA 3-8B | **0.105** | 0.122 | 0.127 | 0.126 | 0.140 |
| Mixtral | **0.087** | 0.161 | 0.145 | 0.135 | 0.116 |
| Gemma 2 | **0.107** | 0.173 | – | – | – |

Table 5: Hellinger distance values between LLM models and optimal models, averaged across Trajectories 1–10, for each model configuration. **Bold** indicates model(s) with the lowest distance metric for the given LLM.

| Model | LAIP (Full) | LAIP (LCP) | LAIP (CoT) | Generic CoT | Baseline |
|---|---|---|---|---|---|
| GPT-4o | 0.118 | **0.100** | 0.191 | 0.324 | 0.317 |
| GPT-4o mini | 0.146 | **0.139** | 0.264 | 0.479 | 0.475 |
| GPT-3.5 | **0.339** | 0.466 | 0.523 | 0.494 | 0.469 |
| LLaMA 3-70B | **0.259** | 0.279 | 0.385 | 0.426 | 0.384 |
| LLaMA 3-8B | **0.309** | 0.355 | 0.357 | 0.352 | 0.378 |
| Mixtral | **0.292** | 0.404 | 0.380 | 0.370 | 0.347 |
| Gemma 2 | **0.312** | 0.425 | – | – | – |

### A.4.1 LAIP MODEL SIZE RESULTS

**Comparing LAIP-Full to LAIP-LLM computes posterior**

GPT-4o: Cohen's $d = -0.03$, $t(14) = -0.05$, $p = .96$

GPT-4o-mini: Cohen's $d = 0.21$, $t(14) = 0.42$, $p = .68$

GPT-3.5: Cohen's $d = 1.04$, $t(14) = 2.09$, $p = .056$

LLaMA 3-70B: Cohen's $d = 0.63$, $t(14) = 1.25$, $p = .23$

LLaMA 3-8B: Cohen's $d = 0.85$, $t(14) = 1.25$, $p = .11$

Mixtral: Cohen's $d = 1.10$, $t(14) = 2.19$, $p = .046$

Gemma 2: Cohen's $d = 1.59$, $t(14) = 3.17$, $p = .007$

**Comparing LAIP-Full to Zero-Shot Baseline**

GPT-4o: Cohen's $d = 1.42$, $t(14) = 2.84$, $p = .013$

GPT-4o-mini: Cohen's $d = 2.93$, $t(14) = 5.86$, $p < .001$

GPT-3.5: Cohen's $d = 1.74$, $t(14) = 3.50$, $p = .003$

LLaMA 3-70B: Cohen's $d = 2.46$, $t(14) = 4.92$, $p < .001$

LLaMA 3-8B: Cohen's $d = 1.57$, $t(14) = 3.15$, $p = .007$

Mixtral: Cohen's $d = 1.30$, $t(14) = 2.61$, $p = .020$

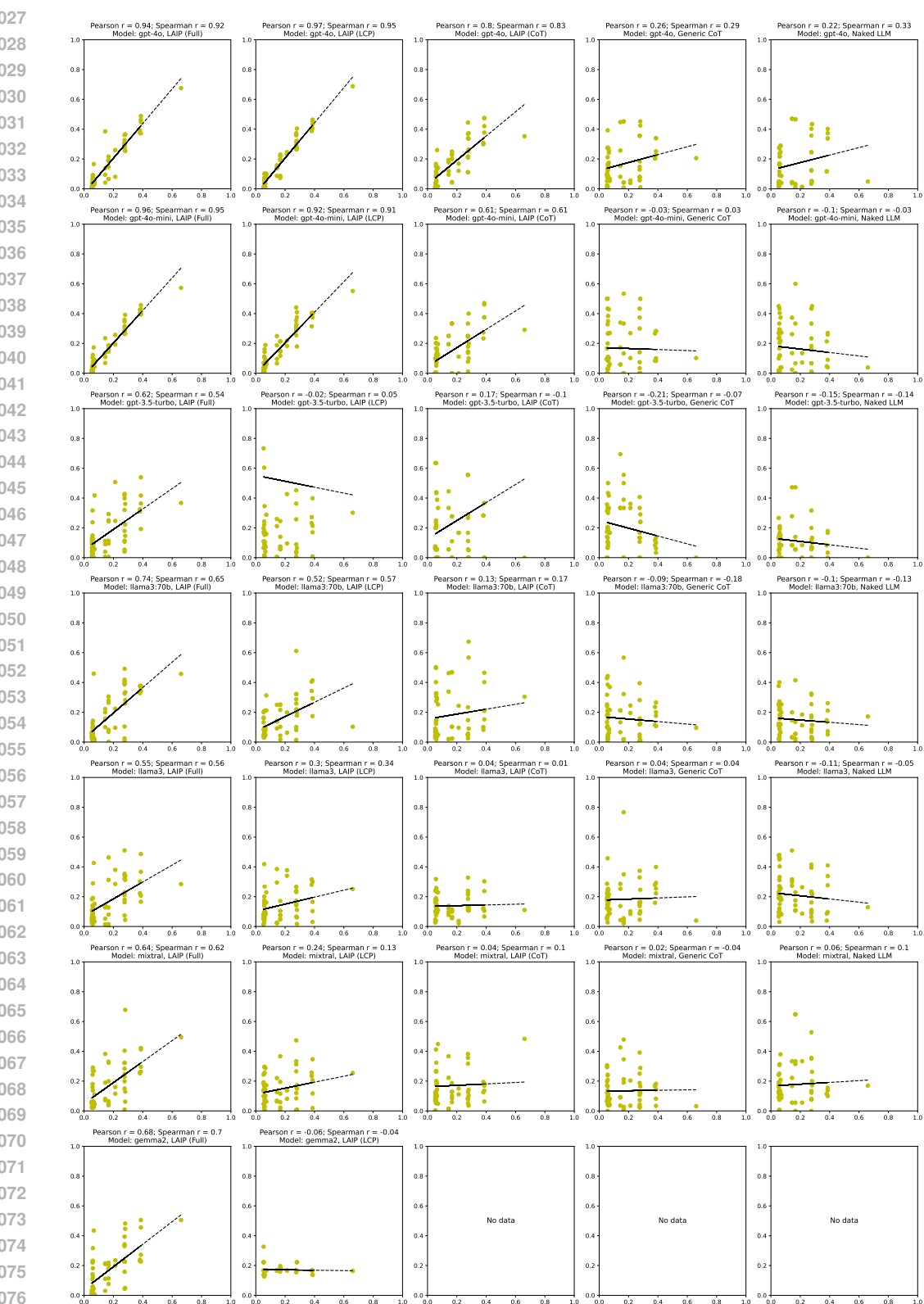

Figure 9: Correlations between optimal model and LLMs, by study and condition.

A.5 UNCONSTRAINED ACTION SPACE: STUDY DETAILS

---

**Situation:**

You are Carol, a coworker of Alice's. Alice is having a surprise birthday party in a few weeks, and it is your job to book a restaurant. Because you are her coworker, you often see her eat lunch, so you are trying to determine what Alice's favourite foods are in order to book the right restaurant.

Here are some things that you already know:

- In the food court at the building where you work, there is a coffee shop, a pizza place, a sushi place, a burger place, a shawarma place, and a sandwich place.

- You do not know whether Alice likes or dislikes any of these, but you know she has been to the food court before.

- Alice might like a cuisine or food that is not listed here.

**Scenes:**

- Today, you are in the food court. You are not feeling especially hungry, but it is lunchtime, and the topic of conversation has come up about where the two of you should eat.

- Today, you are working on a project downtown, and it's time for lunch. You are very hungry. There are many global options to choose from, and you are near a neighbourhood with lots of regional Chinese options. There are also some restaurants serving Thai and Malaysian food a bit further afield, as well as the usual fast food options, like burgers, pizza, and fries.

- Today, you are out of town on a work trip. It is the middle of the day, and you are in a very small town with very few options for something to eat. Looking at your phone, you see that the only options are some small American-style fast food restaurants and some shops with coffee and donuts.

- Today, Alice and Carol have a day off from work. They are not at work, so there are a lot of restaurant options to choose from around the world in the neighbourhood. There is also the option of staying at home and making something from Alice's pantry. However, Alice is clearly feeling very sick, and needs something plain to settle her stomach.

---

Figure 10: Situation prompt for Unconstrained Action Space scenario

Table 6: Posterior probability of LAIP model and Zero-shot Baseline model in unconstrained action space task.

| Description | Posterior Probability | |
|---|---|---|
| | LAIP | Zero-shot Baseline |
| Alice loves comfort food: Think mac & cheese, lasagna, hearty stews. | 0.008143 | 0.129368 |
| Alice is a health-conscious eater: She favors salads, lean proteins, and whole grains. | 0.022759 | 0.017936 |
| Alice is a foodie: She enjoys trying new and exotic cuisines. | 0.049308 | 0.140667 |
| Alice is budget-conscious: She prefers affordable and filling meals. | 0.011919 | 0.251269 |
| Alice is picky: She has very specific tastes and dislikes many common foods. | 0.027189 | 0.024988 |
| Alice's favorite cuisine is Italian: Pizza, pasta, and gelato are her go-to's. | 0.005981 | 0.002422 |
| Alice is obsessed with Japanese food: Sushi, ramen, and tempura are her favorites. | 0.008363 | 0.002422 |
| Alice loves Mexican food: Tacos, burritos, and enchiladas are her weakness. | 0.145349 | 0.002422 |
| Alice craves Indian food: Curries, naan bread, and samosas are her favorites. | 0.185195 | 0.002422 |
| Alice is a Thai food enthusiast: Pad Thai, green curry, and spring rolls are her go-to's. | 0.188543 | 0.023245 |
| Alice grew up eating Chinese food: She has a fondness for dim sum, stir-fries, and noodles. | 0.051107 | 0.037266 |
| Alice's family is from Greece: She loves souvlaki, gyros, and baklava. | 0.061148 | 0.002422 |
| Alice has a connection to Middle Eastern food: Hummus, falafel, and shawarma are her favorites. | 0.068596 | 0.087705 |
| Alice loves seafood: She enjoys anything from oysters to lobster. | 0.014397 | 0.002422 |
| Alice is a vegetarian: She prefers plant-based dishes and avoids meat. | 0.023489 | 0.009225 |
| Alice has a sweet tooth: She loves desserts and pastries. | 0.018313 | 0.006205 |
| Alice is a spicy food fanatic: She loves anything with a kick. | 0.037389 | 0.003103 |
| Alice has a hidden love for breakfast food: Pancakes, waffles, and omelets are her favorites. | 0.006797 | 0.008329 |
| Alice prefers simple, home-cooked meals: She enjoys comfort food classics. | 0.009457 | 0.241314 |
| Alice's favorite cuisine is something completely unexpected and unknown to you. | 0.056560 | 0.004845 |

