# OpenReview forum: "Towards Machine Theory of Mind with Large Language Model-Augmented Inverse Planning"
_ICLR.cc/2025/Conference — Submitted to ICLR 2025_

### Official Review · Reviewer_HwkP · 2024-10-25

**Soundness:** 3
**Presentation:** 3
**Contribution:** 2
**Rating:** 5
**Confidence:** 4

**Summary:**

The authors propose a hybrid approach to machine Theory of Mind (ToM) that leverages large language models (LLMs) to generate hypotheses and likelihood functions within a Bayesian inverse planning framework, which then computes posterior probabilities for an agent’s likely mental states based on its actions. This model is intended to improve upon LLMs alone, which can be unreliable in certain reasoning tasks, as well as traditional Bayesian models, which often struggle with complex scenarios. The combined model, termed LLM-Augmented Inverse Planning (LAIP), outperforms either approach independently. It accurately predicts mental states in the experiments, even when using smaller LLMs that typically struggle in these scenarios.

**Strengths:**

1. This paper introduces a hybrid model, called LLM-Augmented Inverse Planning (LAIP), that combines Bayesian inverse planning with large language models (LLMs). This approach enhances Theory of Mind (ToM) reasoning by using LLMs to generate a wider range of hypotheses and actions than traditional Bayesian methods typically allow.
2. The effectiveness of the LAIP model is tested in controlled experimental settings. By using a restricted action and hypothesis space, the study enables a direct, systematic comparison between LAIP’s predictions and those of a Bayes-optimal model, providing clear insights into the model's strengths and limitations in inferring agent beliefs and preferences. The authors provide a detailed analysis of the trajectories and correlations between the optimal model and LLMs.
3. LAIP also demonstrates adaptability in ambiguous scenarios by dynamically generating and updating multiple hypotheses and possible actions based on new cues. This feature allows the model to attribute mental states even in complex situations where intentions are unclear, supporting more flexible social reasoning.

**Weaknesses:**

1. The authors overlook the discussion and comparison of a closely related method presented in the paper "MMToM-QA: Multimodal Theory of Mind Question Answering." This work also employs language models for scalable Bayesian inverse planning, generating likelihoods of different actions under various hypotheses in a similar way. It also concludes that Bayesian inverse planning with smaller language models can outperform larger models, albeit on different tasks.

2. The authors may consider including additional baselines, such as Sim-ToM [2], SymbolicToM [3], along with similar methods like BIP-ALM [1] and LIMP [4]. These approaches have shown significant improvements in LLM reasoning on Theory of Mind benchmarks.

3. Although the experimental setup follows a classic design, it appears somewhat simplistic and limited. The proposed method aims to serve as a general approach to machine Theory of Mind, so it should be tested in more complex, real-life-like scenarios that involve inferring different human mental states (e.g., beliefs, goals, plans, desires). It would be beneficial if the authors tested their method on at least one additional ToM benchmark to further demonstrate its effectiveness.

References:
1. MMToM-QA: Multimodal Theory of Mind Question Answering, Jin et al, 2024
2. Think Twice: Perspective-Taking Improves Large Language Models' Theory-of-Mind Capabilities, Wilf et al, 2023
3. Minding Language Models' (Lack of) Theory of Mind: A Plug-and-Play Multi-Character Belief Tracker, Sclar et al, 2023
4. MuMA-ToM: Multi-modal Multi-Agent Theory of Mind, Shi et al, 2024

**Questions:**

1. The Hypotheses 1 to 20 in the results and Figure 2 are not clearly introduced. Could you provide an explanation of how these were generated and what they represent?

2. Could you provide an error analysis of the proposed method (types of errors, accuracy of each component of the method) to help clarify its limitations?

---

> ### Author Response · Authors · 2024-11-24
> **Response to reviewer HwkP**
>
> Dear Reviewer HwkP,
>
> Thank you for your thorough and helpful evaluation of this work. We agree with you that a more thorough evaluation of LAIP is merited, and we respond to you point by point below.
>
> > **The authors overlook the discussion and comparison of a closely related method** presented in the paper "MMToM-QA: Multimodal Theory of Mind Question Answering." This work also employs language models for scalable Bayesian inverse planning, generating likelihoods of different actions under various hypotheses in a similar way. It also concludes that Bayesian inverse planning with smaller language models can outperform larger models, albeit on different tasks.
>
> Thank you. We agree that the method and the technique employed in Jin et al. (2024) is very relevant to our discussion, and will discuss this work in greater detail as a relevant piece of related work.
>
> > **The authors may consider including additional baselines,** such as Sim-ToM [2], SymbolicToM [3], along with similar methods like BIP-ALM [1] and LIMP [4]. These approaches have shown significant improvements in LLM reasoning on Theory of Mind benchmarks.
>
> We agree that including additional baseline methods to compare our performance against would fortify the evidence we provide. Given our current experiments focus on tracking agents’ goals across time in text-based contexts, we feel that Sim-ToM and Symbolic-ToM, which primarily track agents’ true, false, or second-order false beliefs, are less well-suited as baselines for our current experiments. Likewise, MuMA-ToM only accepts multimodal input. However, in our response to **Reviewer Xehq**, we show preliminary results comparing our results to ReAct, but we also plan to include BIP-ALM as a baseline comparison in addition to Reflexion.
>
> > Although the experimental setup follows a classic design, it appears somewhat simplistic and limited. The proposed method aims to serve as a general approach to machine Theory of Mind, so it should be tested in more complex, real-life-like scenarios that involve inferring different human mental states (e.g., beliefs, goals, plans, desires). It would be beneficial if the authors tested their method on at least one additional ToM benchmark to further demonstrate its effectiveness.
>
> We agree with the reviewer that a more complex ToM benchmark would serve as a more rigorous test of the model. To that end, we have adapted LAIP to the MMToM-QA benchmark (Jin et al., 2024). Our preliminary results indicate that LAIP is well-suited to this task, placing 94.8% of the probability mass on the correct answer on a task in the “goal given true belief” condition. However, we plan to run LAIP on the full suite of results in order to allow us to more directly compare it to other baselines, including BIP-ALM.
>
> [cont...]

---

> > ### Author Response · Authors · 2024-11-24
> >
> > > **The Hypotheses 1 to 20 in the results and Figure 2 are not clearly introduced.** Could you provide an explanation of how these were generated and what they represent?
> >
> > Hypotheses 1 through 20 were generated using GPT-4o. We used the following system message to provide the LLM with context:
> >
> > ```
> > You are observing a person's actions and trying to determine how much the person likes three types of food:
> > Japanese, Chinese and Mexican.
> > This is the set of rules to help you determine the preferences:
> >
> > There are seven rooms:
> > - Room 1 connects to Room 2.
> > - Room 2 connects to Room 1, Room 3, and Room 4.
> > - Room 3 has a Chinese restaurant in it.
> > - Room 4 is connected to Room 2, Room 5, and Room 6.
> > - Room 5 has a Mexican restaurant in it.
> > - Room 6 connects to Room 4 and Room 7.
> > - Room 7 has a Japanese restaurant in it.
> >
> > Restaurants are visible from different rooms:
> > - The Chinese restaurant is **visible** from Room 1, Room 2, and Room 4.
> > - The Chinese restaurant is **not visible** from Room 6.
> > - The Mexican restaurant is **visible** from Room 2, Room 4, and Room 6.
> > - The Mexican restaurant is **not visible** from Room 1.
> > - The Japanese restaurant is **visible** from Room 4 and Room 6.
> > - The Japanese restaurant is **not visible** from Room 1 or Room 2.
> >
> > The agent knows for sure if a restaurant is open if it gets close enough to it.
> > Each restaurant is almost always open, but sometimes is closed.
> > If the restaurant is closed, it will not open up later.
> > Agents cannot eat at restaurants that are closed, even if they like a food.
> > ```
> >
> > We then prompted the LLM to generate 20 hypotheses about the person’s preferences using the following prompt:
> >
> > ```
> > Imagine that Bob is at a food court. There are 3 restaurants, and he thinks that they are likely open, but can’t be sure until he gets closer. The options are Japanese food and Mexican food, and Chinese food. Bob will need to walk past the Chinese food to get to the Japanese food and the Mexican food, and will not be able to see if they are actually open until he walks past. Your first task is to consider Bob’s food preferences. Think about the options that Bob has and think about which food he likes best, second best, and third best. Write out a series of hypotheses about his preferences. You can add additional hypotheses that you think may influence his opinions. Write out 20 hypotheses, and try to have them as mutually exclusive as possible and things that you think will direct his actions as he walks through the space. Provide a likelihood for the probability of each hypothesis. You will use his actions to determine which hypothesis is most accurate by watching each step, updating your beliefs about how much each is likely true for Bob.
> > ```
> >
> > In Study 2, we focused on the first six hypotheses generated by the LLM, as these represented six mutually-exclusive preference hypotheses. (We post the 20 hypotheses in the comment below.)
> >
> > > **Could you provide an error analysis of the proposed method** (types of errors, accuracy of each component of the method) to help clarify its limitations?
> >
> > Thank you. If we understand the reviewer correctly, analyzing the errors within each component of the method would involve identifying whether the LLM:
> >
> > 1. Generated implausible action probabilities in a given state for a given hypothesis (e.g., suggested that an agent would move towards a dispreferred option, despite a preferred option being potentially available)
> > 2. Failed to appropriately integrate or compute these action probabilities into the likelihood
> > 3. Failed to appropriately integrate or compute a posterior probability given the likelihood.
> >
> > If this is correct, we anticipate all errors occurring within the full LAIP model occur at the first stage, as the other two are computed mathematically. However, within other versions of the model, including the version in which the LLM computes the posterior and the LAIP with a single CoT prompt, errors in steps 2 and 3 could contribute to the lower performance of the model.
> >
> > [cont...]

---

> > > ### Author Response · Authors · 2024-11-24
> > > **Hypotheses in Study 1**
> > >
> > > H1: Bob prefers Japanese food the most, then Mexican, then Chinese.
> > >
> > > H2: Bob prefers Japanese food the most, then Chinese, then Mexican.
> > >
> > > H3: Bob prefers Mexican food the most, then Japanese, then Chinese.
> > >
> > > H4: Bob prefers Mexican food the most, then Chinese, then Japanese.
> > >
> > > H5: Bob prefers Chinese food the most, then Japanese, then Mexican.
> > >
> > > H6: Bob prefers Chinese food the most, then Mexican, then Japanese.
> > >
> > > H7: Bob prefers whichever restaurant is open first (i.e., prioritizes availability over food type).
> > >
> > > H8: Bob prefers whichever restaurant has the shortest line, regardless of the food type.
> > >
> > > H9: Bob prefers whichever restaurant smells best to him at the time, regardless of the food type.
> > >
> > > H10: Bob prefers to try something new and will prioritize a restaurant he hasn’t visited recently.
> > >
> > > H11: Bob will make a random choice between the three restaurants when he gets close enough.
> > >
> > > H12: Bob is craving something filling and perceives Mexican food to be the most satisfying, followed by Chinese, then Japanese.
> > >
> > > H13: Bob is on a budget and will choose the cheapest option, which he perceives to be Chinese food, followed by Mexican, then Japanese.
> > >
> > > H14: Bob prefers food that he perceives to be the healthiest, and in this context, he believes Japanese food is the healthiest, followed by Mexican, then Chinese.
> > >
> > > H15: Bob is looking for comfort food and perceives Chinese food to be the most comforting, followed by Mexican, then Japanese.
> > >
> > > H16: Bob has a time constraint and will choose the restaurant he believes will serve him the quickest, which he thinks is Chinese, followed by Mexican, then Japanese.
> > >
> > > H17: Bob is feeling adventurous and wants something with a bold flavor, so he will prefer Mexican food, followed by Japanese, then Chinese.
> > >
> > > H18: Bob will make his decision based on what he plans to do after eating; he will choose the lightest meal (Japanese) if he has an active afternoon ahead, and the heaviest meal (Chinese) if he plans to relax afterward.
> > >
> > > H19: Bob is influenced by recommendations from others and recently heard positive reviews about the Mexican food, which has made him more inclined to choose it, followed by Chinese, then Japanese.
> > >
> > > H20: Bob is not feeling particularly picky and will choose based on convenience. If all restaurants are open, he will pick whichever one he sees first (Chinese).

---

> ### Author Response · Authors · 2024-11-29
>
> Dear Reviewer HwkP,
>
> Thank you again for your review. We have written **[our general response here](https://openreview.net/forum?id=Y2cGisOWPZ&noteId=iF4mpr08oE)** where we outline the changes to our updated manuscript. We also summarize below the relevant changes we have made to address your concerns.
>
> We expanded our discussion of the related work to include several additional works, including BIP-ALM. Additionally, in order to address your concerns about limited baselines and benchmarks, we added four additional baselines to Study 1, and we tested LAIP against the goal inference questions of the MMToM-QA benchmark, which we found outperformed other models, including BIP-ALM, Sim-ToM, and Symbolic-ToM.
>
> Do you think that these changes are sufficient to improve your evaluation of the paper?

---

### Official Review · Reviewer_Gb3Q · 2024-11-01

**Soundness:** 4
**Presentation:** 3
**Contribution:** 3
**Rating:** 8
**Confidence:** 3

**Summary:**

The authors present an approach called LAIP (LLM-Augmented Inverse Planning) that leverages large language models for Bayesian inverse planning over unconstrained action spaces (i.e. strings of natural language). LAIP prompts an LLM to generate various hypotheses in natural language for a given environment, generates actions in natural language conditioned on a given state of the environment, and evaluates the likelihood of each action conditioned on each hypothesis. These generated actions are then compared with ground-truth actions taken by an agent in the environment, and the posterior distribution of hypotheses is calculated. The authors assess LAIP in two studies and compare various modified LAIP models with baselines including an optimal Bayesian inference model, also assessing LAIP's capabilities when using various SOTA LLMs including GPT models, LLaMA models, and Mixtral.

**Strengths:**

- The proposed approach is completely novel, to my knowledge, and faithfully integrates LLMs as tools for Bayesian inverse planning.
- The proposed approach is assessed in multiple clear studies and against multiple methodological variants (e.g. Single CoT), and using multiple SOTA LLMs as the engine for their approach.
- The prose is very easy to understand, arguments are well-motivated, the background literature is up to date.
- Results are compelling.

**Weaknesses:**

- The paper could use some diagrams to better capture the intuition of inverse planning, perhaps by including an example of agent paths through the restaurant environment in Figure 1.
- The paper could use some diagrams to better capture the pipeline of different LAIP variations, or at least the main LAIP approach, including the main components of the pipeline (the LLM, the hypotheses, the generated actions, the true grounded actions, the true hypothesis) and how they are elicited.
- Some discussion about previous work in using likelihood estimates of strings of text generated by LLMs would be welcomed so that the reader can better contextualize how LLMs have been/can be used for such tasks.
- While this is not exactly a weakness, I think there are some interesting philosophical questions about how LLMs are able to capture these kinds of causal relationships in strings of text that are worthy of discussion. Also questions about how a visually grounded VLM might behave differently than a text-only LLM.

**Questions:**

No questions, interesting paper.

---

> ### Author Response · Authors · 2024-11-24
> **Response to reviewer Gb3Q**
>
> Dear Reviewer Gb3Q,
>
> Thank you for your supportive evaluation of our submission. We will address your comments point by point below.
>
> > **The paper could use some diagrams to better capture the intuition of inverse planning,** perhaps by including an example of agent paths through the restaurant environment in Figure 1.
>
> We agree that a clearer version of Figure 1 would help readers understand the task more clearly, and plan to overhaul the figure to include a sample task trajectory and example visibility for an agent based on its position.
>
> > **The paper could use some diagrams to better capture the pipeline of different LAIP variations,** or at least the main LAIP approach, including the main components of the pipeline (the LLM, the hypotheses, the generated actions, the true grounded actions, the true hypothesis) and how they are elicited.
>
> Thank you, we agree that a larger figure that identifies the pipeline we use for different variations of LAIP could also clarify how these approaches differ from or resemble one another, and to clarify the elicitation of these hypotheses. We will create a graphic that presents the main variants we consider in our work.
>
> > **Some discussion about previous work in using likelihood estimates of strings of text generated by LLMs** would be welcomed so that the reader can better contextualize how LLMs have been/can be used for such tasks.
>
> > While this is not exactly a weakness, I think there are some interesting philosophical questions about how LLMs are able to capture these kinds of causal relationships in strings of text that are worthy of discussion. Also questions about how a visually grounded VLM might behave differently than a text-only LLM.
>
> We agree that these are both important questions. One commonality that we identify across both of these points is that many of these models, including LIMP (Shi et al., 2024), BIP-ALP (Jin et al. (2024), CLIPS (Zhi-Xuan et al., 2024) as well as ours, involve some degree of using LLMs to decompose visual scenes, text-based descriptions of scenes, or other open-ended information into structured, parsed text.
>
> We also agree that some discussion of LLM-generated probabilistic estimates would strengthen the work. Recently, Bradshaw et al. (2024) arXiv:2411.03486v1 [cs.AI] have explored how GPT-4o generates plausible distributional properties when simulating vote shares in both past and future United States elections. Although this is a domain in which there is a high volume of both historical and speculative data that may inform the plausibility of its outputs, it is plausible that various higher-order properties of probability distributions may be represented within LLMs.

---

### Official Review · Reviewer_aqP8 · 2024-11-03

**Soundness:** 3
**Presentation:** 3
**Contribution:** 2
**Rating:** 5
**Confidence:** 5

**Summary:**

In Theory of Mind, many human reason prediction models struggled with the scalability of possible hypotheses and actions. The paper uses LLMs to generate hypotheses and likelihood functions to compute Bayesian inverse planning models. The combination of prior computation with Bayesian models and LLMs mitigates the brittleness often seen in LLM-only approaches.

Initially, the LLM generates a prior belief over possible hypotheses regarding the agent’s preferences. Then for each of the steps it observes the agent’s trajectories and generates reasoning about the agent’s likely choices given the state, and the likelihood of different possible actions correspondingly. LLM updates the posterior distribution over hypotheses after the agent acts.

The experiments are performed on simple ToM tasks. The first experimental scenario is inferring preferred restaurants. There are three restaurants where the agent moves about 5 steps. The second scenario is inferring food preferences with actions on what food to choose. The two scenarios are about the same scale. The paper involves experiments with various kinds of common LLM models, and for each of the experiments LLM generates 20 hypotheses. The proposed method performed better than pure LLM and baselines.

**Strengths:**

The LLM effectively generates a wide range of hypotheses, significantly reducing the computational burden of hypothesis generation. It is also more reliable when combined with the Bayesian model so that the accuracy could be higher than using pure  LLM. LLMs like GPT-4o, which are strong in logical computation, can compute posterior distributions directly, further reducing computational costs. This saves lots of manpower and computation resources.

**Weaknesses:**

Although overall the approach is interesting, there are several weaknesses.

1) The benchmarks provided seem insufficient for a comprehensive evaluation. Currently the comparison has been made with LAIP(LLM computes posterior), LAIP (single LLM call), pure LLM, and a steadily updated model. They are the variation or downgraded versions of the proposed LAIP model. They are good baselines though.

2) The paper claims that the scalability issue is a big problem, and I fully agree. This is exponentially more difficult to calculate the bayesian probability when the state space gets a little larger. That hinders the development of inverse reinforcement learning from this certain direction. However, the paper’s chosen scenarios are simple enough that priors could almost be calculated manually, raising questions about scalability to more complex settings. It can hardly convince that LLM can help solve it as the task might not be too challenging for gpt4o.

3) The paper lacks an explanation for how smaller, less reliable LLMs consistently perform well under LAIP. The results shown in Figure 3 indicate that several small LLM models perform much better with LAIP, but there is no mechanism that ensures the models can still contribute.

**Questions:**

1. How is LAIP compared to sota approaches, or pure-math BIRL models? (Ramachandran 2007)
2. What are the computational and scalability limits of this approach in terms of state space and trajectory length?
3. Can you give analysis on how LLM can boost the small LLM models?
4. Figure 4 seems to be drifting, and would you please include its conclusion in the main text?
5. How do you ensure the dataset you generated is not biased, i.e. it is not your choice of distribution that affects the performance of LAIP?

Ramachandran, Deepak, and Eyal Amir. "Bayesian Inverse Reinforcement Learning." IJCAI. Vol. 7. 2007.

**Details Of Ethics Concerns:**

No concerns.

---

> ### Author Response · Authors · 2024-11-24
> **Response to reviewer aqP8**
>
> Dear Reviewer aqP8,
>
> Thank you for your helpful comments and constructive feedback on our work. We believe your comments raise a number of important points, which we address below.
>
> > **The benchmarks provided seem insufficient** for a comprehensive evaluation. Currently the comparison has been made with LAIP(LLM computes posterior), LAIP (single LLM call), pure LLM, and a steadily updated model. They are the variation or downgraded versions of the proposed LAIP model. They are good baselines though.
>
> > **How is LAIP compared to sota approaches,** or pure-math BIRL models? (Ramachandran 2007)
>
> Thank you for raising this point. As we note in our responses to **Reviewer Xehq** and **Reviewer HwkP**, we agree that our work would be strengthened through the inclusion of a broader number of baselines, including other recent and SOTA approaches. We have run preliminary tests of our model against ReAct, and find LAIP substantially outperforms it on our task. Additionally, we plan to test Reflexion (Shinn et al., 2024) and BIP-ALM (Jin et al., 2024) on our task.
>
> > **The paper claims that the scalability issue is a big problem, and I fully agree.** This is exponentially more difficult to calculate the bayesian probability when the state space gets a little larger. That hinders the development of inverse reinforcement learning from this certain direction. However, the paper’s chosen scenarios are simple enough that priors could almost be calculated manually, raising questions about scalability to more complex settings. It can hardly convince that LLM can help solve it as the task might not be too challenging for gpt4o.
>
> > What are the computational and scalability limits of this approach in terms of state space and trajectory length?
>
> We agree that scaling is a potential concern with the LAIP model in more complex environments. We describe in greater detail in our response to **Reviewer Xehq** our perspective on the degree to which we anticipate LAIP can scale in its current form. The primary challenge of the current approach is that if the hypothesis space grows sufficiently large, generating action probabilities across many candidate hypotheses poses a steep computational overhead, especially as the reviewer notes with longer trajectories.
>
> One potential solution to this approach is an approach in which hypotheses are not held static across the entire trajectory length but can be iteratively revised, pruned, and introduced in a manner similar to particle filtering/sequential Monte Carlo approaches. This would allow persistently low-probability hypotheses to be filtered out, such that they would not require simulation when their prior probability is sufficiently low.
>
> Nevertheless, our preliminary experiment with LAIP in the MMToM-QA environment, which has a more open-ended action space than our task and longer trajectories, suggests that LAIP is able to effectively scale to reason about more complex tasks in which the number of objects to track and candidate actions is much larger. As we describe in our response to **Reviewer Xehq**, although preliminary, our result outperforms the average from BIP-ALM. We plan to test across the other conditions of the benchmark to show its effectiveness across a broader set of scenarios and assumptions.
>
> > **The paper lacks an explanation for how smaller, less reliable LLMs consistently perform well under LAIP.** The results shown in Figure 3 indicate that several small LLM models perform much better with LAIP, but there is no mechanism that ensures the models can still contribute.
>
> > Can you give analysis on how LLM can boost the small LLM models?
>
> We thank the reviewer for raising this point. We believe this finding reflects two strengths of the task decomposition that we use to assess action probabilities. When an LLM directly evaluates the probability of a hypothesis across a full trajectory, or computes and updates probabilities within the language model, it is liable to make errors in computation, mathematical reasoning, or application of mathematical functions; these errors are especially likely to be pronounced in smaller language models that may be less expressive. However, by prompting the LLM to solve smaller tasks, such as possible actions in a given state, assuming one of several candidate hypotheses is true, the LLM may be more likely to provide accurate responses, especially if the posterior is mathematically computed rather than computed using an LLM prompt.
>
> [cont...]

---

> > ### Author Response · Authors · 2024-11-24
> >
> > > **Figure 4 seems to be drifting,** and would you please include its conclusion in the main text?
> >
> > Thank you for identifying this issue. We will ensure that we reference this figure and its findings in the main text.
> >
> > > **How do you ensure the dataset you generated is not biased,** i.e. it is not your choice of distribution that affects the performance of LAIP?
> >
> > We chose trajectories within our “food truck” environment that would distinguish between multiple candidate hypotheses to differing degrees, such that success would sometimes involve concentrating probability mass on a single plausible hypothesis, and sometimes involve distributing probability across multiple plausible hypotheses.
> > Nevertheless, we feel that by showing LAIP can succeed on other benchmarks, as we describe above, we will be able to demonstrate its promise on different environments regardless of the distribution of the dataset.

---

> > > ### Comment · Reviewer_aqP8 · 2024-11-25
> > >
> > > Thank you for the detailed responses and the effort in addressing the feedback. I appreciate the authors' willingness to run additional benchmarks and outline future improvements. However, there are still areas where additional clarification or updates to the manuscript would strengthen the work:
> > >
> > > 1. Additional Benchmarks: I appreciate the authors’ agreement to run more benchmarks, as mentioned in the response. However, it is unclear if the manuscript has been updated to include these results. Highlighting new additions in the text (e.g., with a different color) would make it easier to track changes and assess the updates.
> > >
> > > 2. Scalability and Complexity: The proposed solution to scalability challenges, such as filtering hypotheses with low probabilities, looks promising. Including experiments to validate this approach would greatly strengthen the work. While filtering might help address the computational burden, determining appropriate thresholds and effectively ranking hypotheses in larger state spaces remain challenges. Additionally, generating 20 hypotheses is relatively modest; more evidence on handling larger-scale problems would add significant value.
> > >
> > > 3. Small LLMs in LAIP: The explanation of how smaller LLMs can perform well under LAIP is reasonable and provides good intuition. Addressing this quantitatively, such as with analyses or experiments demonstrating the performance boost for smaller models, would make this argument much more convincing.
> > >
> > > Thank you again for your efforts in responding to the feedback. I appreciate the thoughtfulness of your responses and the potential directions outlined for future work. I hope these updates and clarifications will help enhance the impact and understanding of your contribution.

---

> ### Author Response · Authors · 2024-11-29
>
> Dear Reviewer aqP8,
>
> Thank you for your additional remarks. We have written **[our general response here](https://openreview.net/forum?id=Y2cGisOWPZ&noteId=iF4mpr08oE)** where we outline the changes to our updated manuscript; however, we also briefly address your subsequent questions here.
>
> > **Additional Benchmarks:** I appreciate the authors’ agreement to run more benchmarks, as mentioned in the response. However, it is unclear if the manuscript has been updated to include these results. Highlighting new additions in the text (e.g., with a different color) would make it easier to track changes and assess the updates.
>
> On `Lines 412–441`, we detail the results of LAIP on the MMToM-QA benchmark. We find that it outperforms the BIP-ALM model proposed in Jin et al. (2024), as well as other models such as Sim-ToM and Symbolic-ToM, in an average across all goal conditions, and remarkably so in the goal-updated condition. The large action and state space and long trajectories of this benchmark also provide strong evidence of LAIP's ability to handle more complex, challenging problems that may remain difficult for a pure-math IRL approach.
>
> > **Scalability and Complexity:** The proposed solution to scalability challenges, such as filtering hypotheses with low probabilities, looks promising. Including experiments to validate this approach would greatly strengthen the work. While filtering might help address the computational burden, determining appropriate thresholds and effectively ranking hypotheses in larger state spaces remain challenges. Additionally, generating 20 hypotheses is relatively modest; more evidence on handling larger-scale problems would add significant value.
>
> We agree with the reviewer that solving scalability is an important concern. Effectively determining which hypotheses to maintain and which to discard, or when to revise a hypothesis, and how much effort to expend on this process of considering multiple hypotheses is one which we believe to be a challenge, but one that is within the abilities of LLMs. We intend to work on this issue, particularly as it concerns modelling the social or motivational aspects that may underlie when human beings engage in effortful cognition to infer someone else's beliefs, desires, or goals using inverse planning; however, the goal of this work is to first demonstrate the utility of this approach in open-ended environments. While these represent promising avenue for further iteration and improvement, they may be out of scope of this initial paper in which we demonstrate the basic utility of the algorithm.
>
> Nevertheless, we have added several paragraphs in the discussion of the paper (Lines `509–524`) to address this important issue and highlighted techniques that we intend to use to address further environmental complexity in future work, including particle filter-inspired model designs.
>
> > **Small LLMs in LAIP:** The explanation of how smaller LLMs can perform well under LAIP is reasonable and provides good intuition. Addressing this quantitatively, such as with analyses or experiments demonstrating the performance boost for smaller models, would make this argument much more convincing.
>
> Thank you. We have added an analysis comparing the relative improvement between the baseline model between the nested improvements of the model on Study 2, focusing on the difference between the full model and the model where the LLM computes the posterior (Lines `397–406`). We find that large models such as GPT-4o and LLaMA 3-70B show the least improvement, while Gemma and Mixtral, the smallest models, show significant improvement and the largest effect size difference between these two conditions.
>
> > Thank you again for your efforts in responding to the feedback. I appreciate the thoughtfulness of your responses and the potential directions outlined for future work. I hope these updates and clarifications will help enhance the impact and understanding of your contribution.
>
> Thank you for your constructive reviews. We hope that the changes to the manuscript and the comments above outlining these changes address your concerns. Do you feel satisfied that we have done so sufficiently to increase your evaluation?

---

> > ### Comment · Reviewer_aqP8 · 2024-12-01
> >
> > Thank you for your detailed response in this additional round. I appreciate the effort you have put into improving the paper since the initial draft. The manuscript has significantly improved, and your responses have addressed most of my concerns.
> >
> > However, I remain concerned about your statement regarding the problem being challenging for pure mathematical models due to scalability issues. As mentioned in the paper, this particular scalability challenge is not addressed here either. Furthermore, the current scalability limitations do not appear to be inherently prohibitive for pure mathematical models, and there is no empirical or analytical comparison to substantiate this claim.
> >
> > While I am happy to raise my scores based on the improvements made, I still feel that the paper does not meet the threshold for acceptance.

---

> > > ### Author Response · Authors · 2024-12-02
> > >
> > > Thank you for your re-evaluation of our work, and for your thoughtful and helpful feedback throughout.
> > >
> > > We agree that scalability remains an important concern. Nevertheless, there are inherent properties to pure mathematical models that—while being suitable to evaluate problems such as the one our Bayes-optimal model solves in Study 2—are prohibitive to their use in open-ended contexts.
> > >
> > > One of them we mention within the text as representing a "frame problem" for ToM reasoning (or other form of generalized Bayesian reasoning) that utilizes pre-defined hypothesis and action spaces. Since open-ended environments contain a potentially unbounded number of auxiliary variables, within a purely mathematical model, the relationship of each one of these variables to a similarly unbounded action space must be prespecified or learned. Although full Bayesian inference on a problem of this scale is intractable, human beings have strong hierarchical inductive constraints (for some examples, see e.g. [1–4]) that enable people to gain structured knowledge across domains without falling victim to this problem of underdetermination. Bayesian models of these tasks are often quite accurate and useful, given the specific bounds laid out on the tasks they solve (such as by restricting the hypothesis space, the state space, and the inductive constraints placed on these models). The scale of problems that can be solved with pure mathematical methods has been considerably expanded through the use of approximate algorithms such as Monte Carlo-based simulations that approximate a posterior distribution, but typically remain constrained to a single task or family of tasks.
> > >
> > > On the other hand, one of the strengths of deep learning methods broadly construed is their ability to encode high-dimensional knowledge. These methods have also proven to be a promising method in improving the performance of IRL algorithms more generally (e.g., [5, 6]). By encoding large amounts of domain-general associations in the form of latent language-based representations, we believe that hybrid models incorporating LLMs as hypothesis or action space samplers (e.g. [7]) have the potential to take advantage of this in order to reason about high-dimensional spaces which have complex or ill-defined action or state spaces. It is here that we believe pure mathematical models may encounter issues with scalability not only in terms of the number of variables that must be explicitly considered and conditioned on, but also issues with defining the boundaries or operationalization of a variable within a task context.
> > >
> > > Finally, we want to thank you again for the engaging conversation.
> > >
> > > [1] Piantadosi, S. T., Tenenbaum, J. B., & Goodman, N. D. (2016). The Logical Primitives of
> > > Thought: Empirical Foundations for Compositional Cognitive Models. _Psychological Review_.
> > > [doi:10.1037/a0039980](https://doi.org/10.1037/a0039980)
> > >
> > > [2] Tenenbaum, J. B., Kemp, C., Griffiths, T. L., & Goodman, N. D. (2011). How to grow a mind: Statistics, structure, and abstraction. _Science, 331_(6022), 1279-1285. [doi:10.1126/science.1192788](https://doi.org/10.1126/science.1192788)
> > >
> > > [3] Dewar, K. M., & Xu, F. (2010). Induction, overhypothesis, and the origin of abstract knowledge: Evidence from 9-month-old infants. _Psychological Science, 21_(12), 1871-1877. [doi:10.1177/0956797610388810](https://doi.org/10.1177/0956797610388810)
> > >
> > > [4] Gweon, H. (2021). Inferential social learning: Cognitive foundations of human social learning and teaching. _Trends in Cognitive Sciences, 25_(10), 896-910. [doi:10.1016/j.tics.2021.07.008](https://doi.org/10.1016/j.tics.2021.07.008)
> > >
> > > [5] Wulfmeier, M., Ondruska, P., & Posner, I. (2015). Maximum entropy deep inverse reinforcement learning. arXiv:1507.04888 [cs.LG]
> > >
> > > [6] Brown, D. S., & Niekum, S. (2019). Deep bayesian reward learning from preferences. arXiv:1912.04472 [cs.LG]
> > >
> > > [7] Wang, R., Zelikman, E., Poesia, G., Pu, Y., Haber, N., & Goodman, N. D. (2024). Hypothesis search: Inductive reasoning with language models. In _ICLR_.

---

### Official Review · Reviewer_Xehq · 2024-11-04

**Soundness:** 3
**Presentation:** 3
**Contribution:** 3
**Rating:** 6
**Confidence:** 3

**Summary:**

* LAIP is an approach that uses LLMs as components of a cognitive model, specifically the bayesian inverse planning model that has explained human behavior in cognitive science theory of mind tasks
* LAIP uses LLMs to generate hypotheses and likelihood functions, then applies Bayesian inverse planning to compute posterior probabilities over an agent's mental states based on observed actions
* Experiments show that LAIP outperforms LLMs alone and chain-of-thought prompting on hand crafted tasks similar to the Food truck tasks from Baker et al.
* The model also demonstrates some potential (albeit with limited envs) in predicting mental states in more open-ended tasks than Bayesian methods can practically handle, aiming to create socially intelligent generative agents

**Strengths:**

* Smart integration of LLMs with inverse planning—leverages the open-endedness of LLMs and the reasoning strengths of Bayesian models.
* Addresses limitations of both LLMs (brittleness, reasoning errors) and Bayesian models (scaling issues with hypothesis/action spaces).
* Experimental results are promising and cool to see them match Bayesian methods in posteriors
* Potential for application in developing socially intelligent agents and enhancing human-AI interaction

**Weaknesses:**

* Limited baselines: the baselines in the paper do not reflect the current SOTA of these approaches. For Study 1
you should add few shot prompting, CoT, ReAct, Reflexion type baselines as well. Zero shot baseline is not a strong baseline
* Experiments seem narrow in scope—focused on the food truck type toy tasks from cognitive science. The true promise of the method is that it should be more scalable than bayesian methods yet a majority of the experiments were with environments that Bayesian methods could handle. The number of potential hypotheses are often small as well, simplifying the process of latent state inference to a multiple choice problem. It would be interesting to see the methods on open ended ToM inference or multiagent decision-making tasks or benchmarks. Needs testing in diverse, unpredictable environments
* Scalability and computational efficiency aren't thoroughly addressed—how does LAIP perform with more complex tasks?
* Figures, results, and what they are highlighting could be more clear. Especially for a broader ML audience that may not be familiar with these tasks. Figure 1 in particular is confusing

Minor:
* Missing related work: Hypothetical Minds and ProAgent. Similarity to these approaches should be discussed

**Questions:**

How is posterior probability of a hypothesis computed with the LLMs? Can you make this clearer in the main text

What's the computational overhead of combining LLMs with Bayesian inverse planning?

---

> ### Author Response · Authors · 2024-11-24
> **Responses to reviewer Xehq**
>
> Dear Reviewer Xehq,
>
> Thank you for your thoughtful and constructive feedback. We will address your comments and questions point by point.
>
> > **Limited baselines:** the baselines in the paper do not reflect the current SOTA of these approaches. For Study 1 you should add few shot prompting, CoT, ReAct, Reflexion type baselines as well. Zero shot baseline is not a strong baseline
> We agree with you that exhibiting more robust evidence of LAIP’s effectiveness by comparing to existing baselines and other recent approaches is important.
>
> To address this point, we have run several new experiments which we describe below, and we plan to include additional baseline comparisons for Studies 1 and 2, including zero-shot CoT, few-shot CoT using reasoning about a different trajectory, ReAct, and Reflexion, to compare against the predictions of the LAIP model.
>
> For Study 1, we have run additional preliminary results using a generic zero-shot CoT prompt, LAIP with zero-shot CoT prompt, and ReAct, using a generated prior across hypotheses (all using GPT-4). We find that, across an average of 6 runs, LAIP with CoT correctly identifies Hypothesis 2 as the most likely (51.9% probability), albeit less than the full model (77.2% probability). Both of these new experiments substantially outperform the zero-shot CoT baseline (11.6% probability) and ReAct (3.6% probability).
>
> > **Experiments seem narrow in scope**—focused on the food truck type toy tasks from cognitive science. The true promise of the method is that it should be more scalable than bayesian methods yet a majority of the experiments were with environments that Bayesian methods could handle. The number of potential hypotheses are often small as well, simplifying the process of latent state inference to a multiple choice problem. It would be interesting to see the methods on open ended ToM inference or multiagent decision-making tasks or benchmarks. Needs testing in diverse, unpredictable environments
>
> We also agree that showing that LAIP can succeed in more open-ended environments where the hypothesis or action space are less constrained is a valuable addition to this work; indeed, showing that LLM-generated hypothesis and action spaces can be leveraged to successfully reason using inverse planning in open-ended domains is the largest conceptual contribution that we hope to be able to more fully demonstrate.
>
> Following the suggestions of another reviewer, we have adapted the LAIP model to the MMToM-QA benchmark (Jin et al., 2024). This benchmark uses a more complex environment with a larger number of free parameters, in which the observed actor traverses through an apartment with various rooms, each of which contains various containers or surfaces on which various foods, utensils, and household objects are located. On this task, the LLM must reason about the likely beliefs or goals of an actor given their trajectories.
>
> In a preliminary run from the “goal given true belief” condition in MMToM-QA using the LAIP model, we have found LAIP identified the actor’s true goal with 94.8% of the probability mass on the correct answer, compared to 77.3% across all “goal given true belief” tasks for the text-only BIP-ALM model with fine-tuned GPT-J. We will run across additional conditions in order to provide a more comprehensive benchmarking of our model.
>
> > **Scalability and computational efficiency aren't thoroughly addressed**—how does LAIP perform with more complex tasks?
>
> > What's the computational overhead of combining LLMs with Bayesian inverse planning?
>
> We thank the reviewer for raising these important concerns. We anticipate that by including more complex tasks in which the number of actions or hypotheses the model needs to consider is higher, we will provide good evidence that LAIP is capable of scaling in terms of its effectiveness in solving more complex environments.
>
> We also acknowledge that the computational overhead of LAIP can increase considerably, particularly as the number of hypotheses to consider increases to larger amounts. Given that LAIP seems to be particularly useful at improving the performance of smaller models (e.g., LLaMA 3:7B), this might mean that LAIP might enable or facilitate smaller LLMs to perform more effectively on belief, desire, or goal inference tasks, trading off against increased computational overhead.
>
> We will further elaborate on the overhead resulting when the number of plausible candidate hypotheses is very large as a potential limitation of LAIP’s applicability. We will also evaluate potential solutions to this issue; for example, we anticipate combining LAIP with particle filter/sequential Monte Carlo-based approaches to approximate Bayesian inference could allow the model to entertain a small number of hypotheses at a time, eliminating “particles” with low-probability hypotheses and allowing for new hypotheses to be proposed from a prior across e.g., other relevant features of the current state of the environment.
>
> [cont...]

---

> > ### Author Response · Authors · 2024-11-24
> >
> > Importantly, however, our best performing model computes the posterior mathematically through direct Python calls, resulting in lower overhead than the model in which the LLM computes the posterior. We feel that “hybrid architectures” such as these can minimize the costs of direct LLM calls; although there are costs associated as the complexity of the hypothesis space increases, these costs may be necessary to solve inverse planning problems in domains of corresponding complexity.
> >
> > As with human beings, who may engage in “resource-rational” allocation of cognitive resources, trading off the anticipated benefits of a more complex strategy against the cognitive costs of using it, using LLMs to reason about inverse planning may be most cost-effective when the demands of a complex environment necessitate the costs of a more resource-intensive strategy. Similarly, optimizing the computational cost of inverse planning may involve a degree of meta-reasoning about the complexity of the task, which can inform the number of hypotheses or the granularity of the action spaces to consider for the task.
> >
> > > **Figures, results, and what they are highlighting could be more clear.** Especially for a broader ML audience that may not be familiar with these tasks. Figure 1 in particular is confusing
> >
> > We appreciate the reviewer raising this concern. We plan to overhaul Figure 1 in particular, to present the environment and the actor’s trajectories in greater detail, as **Reviewer Gb3Q** suggests.
> >
> > > **Minor: Missing related work**: Hypothetical Minds and ProAgent. Similarity to these approaches should be discussed
> >
> > Thank you, we will discuss these approaches in greater detail in the main text, particularly as it concerns their use of multi-agent environments where the reasoning of one agent about another agent’s beliefs, goals, and desires can be directly compared against the ground truth of that agent’s beliefs, goals, and desires.
> >
> > > **How is posterior probability of a hypothesis computed with the LLMs?** Can you make this clearer in the main text
> >
> > We will clarify this in the main text. In the full model, we use regex searching to identify the probability of taking each candidate action given each hypothesis. We then use the column of this matrix corresponding to the action chosen by the actor to derive the likelihood, multiply the prior by the likelihood, yielding an unnormalized probability distribution which we then normalize. When the posterior is computed by the LLM, we provide the LLM with the matrix of all action probabilities given each hypothesis, as well as the corresponding actions each column corresponds to and the hypotheses each row corresponds to, and prompt the LLM to compute the posterior distribution.

---

> > > ### Comment · Reviewer_Xehq · 2024-11-25
> > > **Response to Official Comments**
> > >
> > > I appreciate the detailed responses to my questions and concerns. The new experiments seem promising (albeit preliminary). It would be good to look at them in more detail. Have you integrated these new results and the other changes you mentioned into a new manuscript? If so, please upload it and signify what you have changed in a global response

---

> > > > ### Author Response · Authors · 2024-11-29
> > > >
> > > > Dear Reviewer Xehq,
> > > >
> > > > Thank you. We have written **[our general response here](https://openreview.net/forum?id=Y2cGisOWPZ&noteId=iF4mpr08oE)** where we outline the changes to our updated manuscript.
> > > >
> > > > To address your specific concerns, we added four additional baseline measures for Study 1, including CoT, single-prompt LAIP as CoT, ReAct, and Reflexion. We have also tested our model against the MMToM-QA benchmark, which has a substantially more complex action and state space, showing the model continues to perform well in more complex environments, and in fact outperforms comparable models. Lastly, we have added additional discussion of related work, including Hypothetical Minds and ProAgent, and we have added several paragraphs in the discussion about the computational costs of our procedure and how differing degrees of model complexity could mirror human effort tradeoffs.
> > > >
> > > > Do you feel that these changes address your concerns sufficiently to raise your score?

---

> > > > > ### Comment · Reviewer_Xehq · 2024-12-03
> > > > > **Response**
> > > > >
> > > > > Thank you for addressing my feedback and uploading the new manuscript. I appreciate the additional work you put into enhancing the manuscript. The inclusion of MMToM-QA and CoT, single-prompt LAIP as CoT, ReAct, and Reflexion as baseline measures in Study 1 definitely strengthen the results. I raised my score accordingly.

---

### Author Response · Authors · 2024-11-28
**General Response to Reviewers**

We once again thank the reviewers for their helpful comments and constructive suggestions, which we believe have substantially improved the updated paper that we have posted. We have used the review period to test several new baselines against our task and adapt our model to run on the MMToM-QA benchmark, which has various challenging goal inference scenarios. We have also created several new figures to clarify the model and task designs, included several new analyses of our data, and expanded our discussion of related work and of potential avenues for advancement.

In their initial comments, reviewers identified a number of areas of concern or with potential for improvement. We will summarize these into a few general areas, and outline how we have updated our draft to integrate these improvements and address these concerns.

### Related Work

We have added brief discussion of various other ToM models leveraging LLMs, including Hypothetical Minds, MuMA-ToM, ProAgent, Sim-ToM, and BIP-ALM, and discussed their contributions regarding multi-agent and multi-modal ToM tasks (Lines `110–127`).

### Figure Clarity

**Reviewers Xehq** and **Gb3Q** requested that figures be created or redesigned to more clearly communicate the task and the model design. On Lines `162–180` and Lines `216–229`, we have substantially reworked the original Figure 1 (current Figure 2) of the actor trajectory on our task, and the new Figure 1 presents a graphical schematic of the LAIP model. We have moved the pseudocode for LAIP to the appendix, as its purpose now overlaps with Figure 1.

### Limited Baselines

**We have added several new baselines to Study 1.** In addition to including a generic CoT prompt, as well as a version of the LAIP model with the instructions for the model as a single CoT prompt, we have added results from ReAct (Dagan et al., 2023) and Reflexion (Shinn et al., 2024) on our task.

As we show on our new graph on Lines `240–264`, only the full LAIP model and the LAIP-CoT model recognize the behaviour on the trajectory as reflecting a preference for the Japanese restaurant when the Japanese restaurant is closed, while both ReAct and Reflexion indicate a strong preference for the Chinese restaurant (the one ultimately chosen by the actor). Our full model was consistently the most successful on the task.

### Stronger test of the model / Benchmarking against existing datasets

We tested our model against the goal inference tasks of the MMToM-QA dataset (Jin et al., 2024), a task that closely matched ours in terms of design, but with a larger action space, environmental complexity, and requiring reasoning about goals given true or false beliefs, unrealized goals, and updated goals. This demanding task has especially found consistently low performance for text-only models on the “updated goal” task, including 4.2% for GPT-4, 2.7% for Sim-ToM, 0.0% for Symbolic-ToM, and 30.7% for BIP-ALM with finetuned GPT-J, and no text model performing better than 50.7%.

**With LAIP, we achieve 80.4% success on this task, notably outperforming not only other text models but also all other multimodal models as well**. Our overall results also outperform other text models. A summary of our results and comparison with other models has been added to Lines `412–441`.

### Scalability and Computational Efficiency

Several of the reviewers discussed concerns with the scalability of the approach. We have now moved the pseudocode to the appendix, but our approach, while potentially more computationally intensive as the number of hypotheses increases, scales linearly with the number of hypotheses to consider and the length of an episode. Nevertheless, we discuss the limitations to this approach, as well as the relevance of this topic to the field of “resource-rational” models of human cognition and potential solutions inspired by this domain, in the Discussion on lines `509–524`.

As we elaborate upon in our discussion of Study 2, one remarkable finding we show is that LAIP allows even very small models such as LLaMA 3-8B, Gemma 2, and Mixtral to outperform GPT-4 at baseline and GPT-4 with a zero-shot CoT. Given the large differences in model size and parameters between these models, we believe this finding, combined with our new analyses showing that these small models benefit especially strongly from LAIP (Lines `397–406`), illustrate the benefits that LAIP may provide. This feature may make it possible to leverage more powerful LLM tools, even when hardware does not permit using very large models on a given system.

---

### Meta-Review · Area_Chair_f1gQ · 2024-12-19

**Metareview:**

This paper introduces a method that combines language models and Bayesian inverse planning for Theory of Mind reasoning. The proposed hybrid approach leverages an LLM to propose hypotheses and evaluate action likelihood given hypotheses for conducting Bayesian inverse planning. Experiments on limited settings showed promising results. After the rebuttal and reviewer discussion, there are still two remaining concerns:

1. **Similarity with prior works that also leverage LLM to conduct inverse planning has not been properly addressed.** The new discussion in the revision simply stated "We extend this line of work by designing an explicit inverse planning model that uses an LLM to generate hypotheses and consider the likelihoods of possible actions across a potentially open-ended hypothesis and action space." This doesn't seem to be accurate as the previous methods also explicitly do inverse planning by evaluating action likelihood given hypotheses in an open-ended manner. So it is still unclear how much novelty this method really has compared to prior works.

2. **The new evaluation on MMToM-QA is incomplete and the results are not really that much better than prior works with smaller LMs.** There is no result on the belief inference questions (which is half of the MMToM-QA dataset), and the authors have not provided any explanations. In the general response, they highlighted a really good performance under one subtype. However, after checking the full results, the performance on 2 out of 4 subtypes for the goal inference questions is lower than the prior SOTA which uses GPT-J instead of GPT-4. It is even more concerning to see that the accuracy of goal inference with false belief is below chance. This suggests that LAIP likely has some systematic bias towards certain types of questions. Additionally, prior SOTA on MMToM-QA can work with multimodal inputs too, whereas LAIP only works with text. So it is still unclear what advantage LAIP has over previous SOTA (which critically also uses LLMs to evaluate action likelihood given hypotheses).

Overall, this appears to be an interesting paper (connecting LLMs with cognitive modeling), but the authors haven't properly clarified the similarity with prior works. The evaluation still seems insufficient even with the new experiments. I would encourage the authors to address these two concerns for a future submission.

**Additional Comments On Reviewer Discussion:**

The initial reviews raised two main concerns: similarity to prior works (e.g., Hypothetical Minds, BIP-ALM, and LIMP) and lack of systematic evaluation on common, challenging ToM benchmarks. The authors added some discussion about the prior works and included a partial evaluation of a recent ToM benchmark (MMToM-QA). The final recommendations were diverging. I read all the author-reviewer discussions and the revision. The revision certainly improved the paper, but I think the two main concerns have only been partially addressed.  Please see the metareview for details.

---

### Decision · Program_Chairs · 2025-01-22

Reject